



# A field-based thickness measurement dataset of fallout pyroclastic deposits in the peri-volcanic areas of Campania region (Italy): Statistical combination of different predictions for spatial thickness estimation

Pooria Ebrahimi[1,2], Fabio Matano[1], Vincenzo Amato[3], Raffaele Mattera[4], Germana Scepi[5]

[1] Institute of Marine Sciences (ISMAR), National Research Council (CNR), Naples, 80133 Italy
[2] Department of Earth, Environmental and Resources Sciences, University of Naples Federico II, Naples, 80126, Italy
[3] Department of Biosciences and Territory, University of Molise, Pesche (Isernia), 86090, Italy
[4] Department of Social and Economic Sciences, Sapienza University of Rome, Rome, 00185, Italy
[5] Department of Economics and Statistics, University of Naples Federico II, Naples, 80126, Italy

*Correspondence to*: Fabio Matano (fabio.matano@cnr.it)

**Abstract.** Determining spatial thickness (z) of fallout pyroclastic deposits plays a key role in volcanological studies and shedding light on geomorphological and hydrogeological processes. However, this is a challenging line of research because: (1) field-based measurements are expensive and time-consuming; (2) the ash might have been dispersed in the atmosphere by several volcanic eruptions; and (3) wind characteristics during an eruptive event and soil-forming/denudation processes after ash deposition on the ground surface affect the expected spatial distribution of the fallout pyroclastic deposits. This article tries to bridge this knowledge gap by applying statistical techniques for making representative predictions. First, we compiled a field-based thickness measurement dataset (https://doi.org/10.5281/zenodo.8399487; Matano et al., 2023) of fallout pyroclastic deposits in several municipalities of Campania region, southern Italy. Second, 18 predictor variables were derived mainly from digital elevation models and satellite imageries and assigned to each measurement point. Third, the stepwise regression (STPW) model and random forest (RF) machine learning technique are used for thickness modeling. Fourth, the estimations are compared with those of three models that already exist in the literature. Finally, the statistical combination of different predictions is implemented to develop a less biased model for estimating pyroclastic thickness. The results show that prediction accuracy of RF (RMSE < 82.46 and MAE < 48.36) is better than that existing literature models. Moreover, statistical combination of the predictions obtained from the above-mentioned models through Least Absolute Deviation (LAD) combination approach leads to the most representative thickness estimation (MAE < 45.12) in the study area. The maps for the values estimated by RF and LAD (as the best single model and combination approach, respectively) illustrate that the spatial patterns did not alter significantly, but the estimations by LAD are smaller. This combined approach can help in estimating thickness of fallout pyroclastic deposits in other volcanic regions and in managing geohazards in areas covered with loose pyroclastic materials.

**Keywords**: fallout pyroclastic deposit thickness, random forest, stepwise regression, ensemble estimates, predictive modelling.





## 1. Introduction

A significant quantity of ash is dispersed in the atmosphere during an explosive volcanic eruption which deposits over a large area of ground surface following wind transportation. Spatial thickness of the ash layer typically decreases with distance from the eruptive vent (e.g., see Perrotta and Scarpati, 2003; Albert et al., 2019) and noticeably influences geomorphological and hydrological processes such as landscape evolution, hillslope hydrology, erosion, and slope stability because the geotechnical and hydraulic properties of the unconsolidated ash layer usually differ from the underlying bedrock and soil. A deep understanding of the thickness of fallout pyroclastic deposits could, therefore, help address geohazard management and many related socioeconomic concerns. It is challenging to estimate the spatial thickness of fallout pyroclastic deposits because there might be more than one eruptive event, the ash-dispersal pattern is influenced by the changes in wind characteristics (e.g., the speed and direction of the wind) during a single eruption, and soil-forming and denudation processes continuously influence the expected spatial thickness due to different slope exposures and geometry. Only costly and time-consuming detailed field-based measurements may hallow to effectively map the thickness spatial variations of fallout pyroclastic deposits over a limited area (see for examples Matano et al., 2016, and Cuomo et al., 2021) The spatial thickness of fallout pyroclastic deposits under the influence of hillslope processes, accordingly, remained as a knowledge gap (De Vita et al., 2006).

Estimating the residual regolith has been a common practice (e.g., Saulnier et al., 1997; Saco et al., 2006; Tesfa et al., 2009; Segoni et al., 2013) and the implemented models performed better when developed based on independent variables and applied to a specific site or in limited areas (Del Soldato et al., 2018; Matano et al., 2016). Conventional approaches for predicting pyroclastic thickness primarily rely on geological data, but the significant improvements in the availability of remote sensing data along with the recent advances in recording depositional history of fallout pyroclastic deposits present a unique opportunity to enhance prediction accuracy. Moreover, the statistical literature shows that better results can be achieved by combining estimations derived from different models, which has not been adopted for the objectives of this article to date.

This article explores the integration of a wide range of predictor variables, mainly derived from digital elevation model (DEM) and satellite multispectral images, and machine learning techniques (i.e. stepwise regression and random forest). These approaches identify the most relevant variables and capture non-linear relationships between the predictor variables and pyroclastic thickness values in order to improve prediction accuracy. Combination schemes were then applied to the predictions of these methods and those derived from classical approaches, namely Slope Angle Pyroclastic Thickness (SAPT; De Vita et al., 2006), Geomorphological Pyroclastic Thickness (GPT; Del Soldato et al., 2016) and Slope Exponential Pyroclastic Thickness (SEPT; Del Soldato et al., 2018). Finally, the predictions are validated by field-based measurements of fallout pyroclastic deposits to empirically demonstrate that combining the results of different models provides better thickness predictions for the fallout pyroclastic deposits.

Different sections are briefly introduced here to provide a better insight into content of this article. Section 2 introduces the study area, while Section 3 explains data collection in the field along with the methodology for preparing the predictor variables and for predicting thickness of fallout pyroclastic deposits. In the next section 4, a detailed description of the field-based thickness measurement dataset and of the predictor variables is provided. Section 5 discusses the results and highlights the advantages of using statistical combination in predicting thickness of fallout pyroclastic deposits. The concluding remarks and suggestions for future work are presented in the last section.



## 2. Study area

The area of interest encompasses Campania region and the immediate surroundings in south Italy (Fig. 1). It is bounded on the west by the Tyrrhenian Sea and on the east by the Apennine hilly-mountainous inner land with an altitude of up to 2050 m a.s.l. at Mt. Miletto. The area has a Mediterranean climate with hot, dry summers and moderately cool rainy winters. The mean annual temperature is about 10°C in the mountainous areas and roughly 18°C along the coast. The mean annual rainfall ranges from 700 mm in the eastern part of the region to 1800 mm in the central part of the Apennine mountains (Ducci & Tranfaglia, 2008).

The geological units of Apennine mountains are formed by Triassic to Early Miocene carbonate platform limestones and pelagic basin calcareous-pelitic sequences. They are strongly deformed, mainly thrusted eastward (Bonardi et al., 2009; Doglioni, 1991; Patacca et al., 1990) and uncomfortably covered by Middle Miocene to Pliocene thrust-top basin fillings, formed by siliciclastic sequences (mostly clay, sandstone, and conglomerate) (Di Nocera et al., 2006).

The Quaternary extension in the hinterland and axial sectors of Campania region territory caused several fluvio-lacustrine intramontane basin openings (Ferranti et al., 2005; Amato et al., 2018; Boncio et al., 2022). The NW-SE and NE-SW striking faults delimit the strongly subsiding coastal basins (e.g., Volturno, Campania, Sarno and Sele plains) along the Tyrrhenian belt where the volcanic complexes of Somma-Vesuvius, Phlegrean Fields, Ischia and Roccamonfina occur (Fig. 1). The volcanoes are active (except for Roccamonfina) and erupted at least once in the last 1000 years (Rosi & Sbrana, 1987; Santacroce, 1987). Diffuse degassing, fumaroles and hot springs are observed around and in the submerged sectors of the volcanoes (Rosi & Sbrana, 1987; Chiodini et al., 2001; de Lorenzo et al., 2001). The above-mentioned volcanic complexes are briefly introduced below:

- The Somma–Vesuvius is an asymmetric, polygenic volcanic complex formed by the superimposition of the younger Vesuvius volcanic cone on the older Mt. Somma. Its morphostructure results from the combined action of NW–SE faulting and large caldera collapses in 18 and 79 A.D. (Milia et al., 2012; Passaro et al., 2018).

- The Phlegrean Fields refer to a volcanic field located immediately westward Naples, characterized by a great number of eruptions with a large explosivity index (Fig. 2). The beginning of Phlegrean volcanism is not well constrained yet, but it is well known that activity of the volcanic field started from the super-eruption of the Campanian Ignimbrite (39 ka BP, Deino et al., 1994; De Vivo et al., 2001).

- Ischia Island is the emergent part of a volcanic edifice in the Gulf of Naples, whose activity started before 150 ka BP. The island is composed of the volcanic rocks (mostly trachyte and phonolite) formed by effusive and explosive eruptions, epiclastic deposits and subordinate terrigenous sediments (de Vita et al., 2006a).

- The Roccamonfina volcanic complex was active between 550 and 150 ka BP in the Garigliano river rift valley. It was affected by an intense Plinian activity revealed by very large craters. The central caldera is the result of the eruptive explosions at 353±5 ka BP, while the latest stage of activity featured the edification of the central shoshonitic domes at 150 ka BP (Giannetti, 2001; Rouchon et al., 2008).

The fallout pyroclastic deposits considered in this article are mainly related to Somma–Vesuvius and Phlegrean Fields volcanoes (Fig. 2) because thickness of the Ischia tephra is not considerable on the mainland and the old Roccamonfina deposits have been mostly eroded outside the volcanic edifice. Therefore, only the volcanic history of Somma-Vesuvius and Phlegrean Fields will be further described in this section.

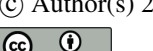



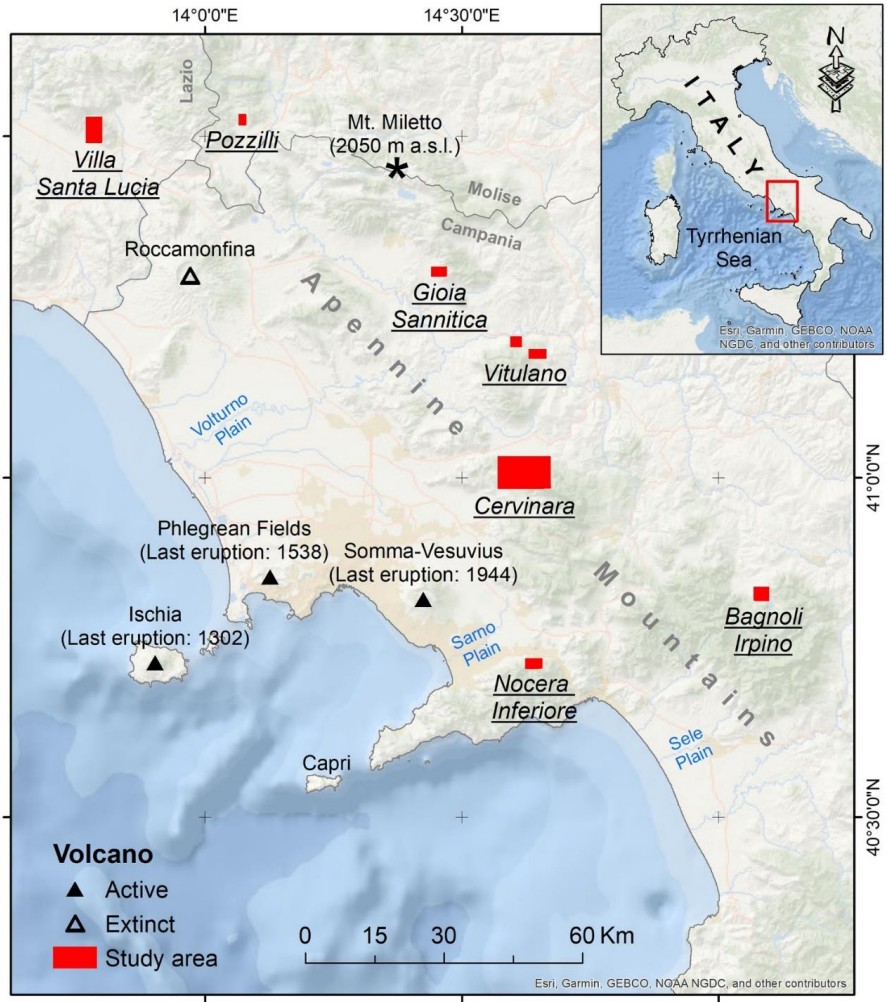

**Figure 1: Location of the study area in southern Italy.**

**2.1. The volcanic history of Phlegrean Fields**

The volcanic activities in Phlegrean Fields started between the late Middle and the early Upper Pleistocene, but the most important ones refer to the Campanian Ignimbrite eruption (CI: 39 ka BP; De Vivo et al., 2001) and the Neapolitan Yellow Tuff eruption (NYT: 15 ka BP, Orsi et al., 1992, 1995; Wohletz et al., 1995; Deino et al., 2004). The former is the most powerful volcanic event ever occurred in the Mediterranean area (Fisher et al., 1993; Orsi et al., 1996; Rosi et al., 1996; Civetta et al., 1997) that emplaced thick sequences of fallout deposits and pyroclastic density currents of mostly trachytic composition, covering an area from the Phlegrean district to Russia (Giaccio et al, 2008; Costa et al, 2022). The outcrops associated with this eruption are found in the Campanian Plain and the Apennine chain (under 80 km from the eruptive vent), mainly composed of a stratified pumice deposit overlay a grey welded tuff unit. In the type sequence, a basal stratified, incoherent ash to sandy deposit and a topmost incoherent coarse pumice deposit with an ashy matrix are also



observed. The NYT eruption mainly emplaced a succession of cineritic and pumiceous lapilli layers (member A) and a deposit made of cinerite layers with dispersed rounded pumices (member B) (Scarpati et al., 1993). The deposits of member B occur as a yellowish massive, lithified tuff in the proximal area, and as an unlithified light grey pumice and ash in the distal outcrops (Cole and Scarpati, 1993; Scarpati et al., 1993).


The post-15ka activity of Phlegrean Fields was concentrated in three epochs separated by two quiescent periods (Fig. 2; Di Vito et al., 1999; Smith et al., 2011; Di Renzo et al., 2011 and references therein), and terminated with the Monte Nuovo eruption in 1538 CE (Guidoboni and Ciuccarelli, 2011; Di Vito et al., 2016 and references therein). The first epoch (15 to ~9.5 ka BP) is characterized by several explosive events, of which Pomici Principali eruption was the most energetic one (Lirer et al., 1987; Di Vito et al., 1999). This epoch was followed by a quiescent period when a thick paleosol layer, pedomarker A, was developed. The second epoch (8.6-8.2 ka BP; Di Vito et al., 1999) is distinguished by only a few episodes of low-magnitude eruptions mainly in NE Campanian Plain. After pedomarker B formation in a prolonged volcanic quiescence, the last epoch of intense volcanic activity began between 4.4 and 3.8 ka BP (Di Vito et al., 1999). The third epoch is characterized by several explosive events, of which the Agnano-Monte Spina eruption (4.4 ka BP; de Vita et al., 1999; Dellino et al., 2001) was the most powerful. This epoch was followed by a prolongate quiescent period and Monte Nuovo eruption (1538 CE; Di Vito et al., 1987; Piochi et al., 2005), respectively. Currently, fumarolic and hydrothermal activities with sporadic bradiseismic episodes mainly occur in Phlegrean Fields. During the 1969-1972 and 1982-1984 bradiseismic crises (Orsi et al., 1999; Del Gaudio et al., 2010; Cannatelli et al., 2020), a total ground uplift of 3.5 m was recorded near the town of Pozzuoli (Barberi et al., 1991). In the last two decades, the central portion of Phlegrean Fields caldera has experienced ground uplift of up to 15 mm/month, and an increase in magnitude and extent of seismicity, especially in 2021-2023 (Falanga et al., 2023).




**2.2. The volcanic history of Somma-Vesuvius**

The Somma-Vesuvius volcanic complex is a few kilometers away from SE Naples. It is composed of the ancient Mt. Somma stratovolcano with a summit caldera in which cone of the internationally recognized Vesuvius volcano was developed. Four major plinian eruptions characterize the Somma-Vesuvius activity (Fig. 2): Pomici di Base (or "Sarno"; Andronico et al., 1995), Mercato (or "Ottaviano"; Rolandi et al., 1993a), Avellino (Rolandi et al., 1993b) and Pompeii (Sigurdsson et al., 1985).


The Pomici di Base eruption is the oldest plinian caldera-forming event, which was followed by notably variable interplinian activities, alternating low-magnitude eccentric flank eruptions, quiescent phases and subplinian events (such as the Greenish Pumice eruption at ~19 ka BP; Santacroce and Sbrana, 2003; Santacroce et al., 2008). About 10 ka later, the Mercato plinian eruption occurred and the products are separated from those of Avellino plinian eruption by a thick paleosol layer (Di Vito et al., 1999). Before the eruption of 79 A.D., the low-intensity eruptions of AP1-AP6 occurred in 3.5-2.3 ka BP (Andronico et al., 2002; Santacroce et al., 2008; Passariello et al., 2010; Di Vito et al., 2019).


The Vesuvius cone was formed by the most recent period of volcanic activity, characterized by a complex alternation of periods of activity with various explosive characters and quiescent phases (Andronico et al., 1995), suddenly interrupted by the Pollena eruption (472 CE; Rolandi et al., 2004). A Middle Age period of variable activity was then started, alternating lava effusions, moderately explosive eruptions, and mild periods (Rolandi et al., 1998), before a subplinian eruption in 1631 CE (Bertagnini et al., 2006). After this event, the volcano entered a semipersistent mild activity with minor lava effusions and short quiescent periods. Each of these periods of repose was preceded by relatively powerful explosive and effusive polyphase eruptions (Arrighi et al., 2001) like the last two ones in 1908 and 1944.





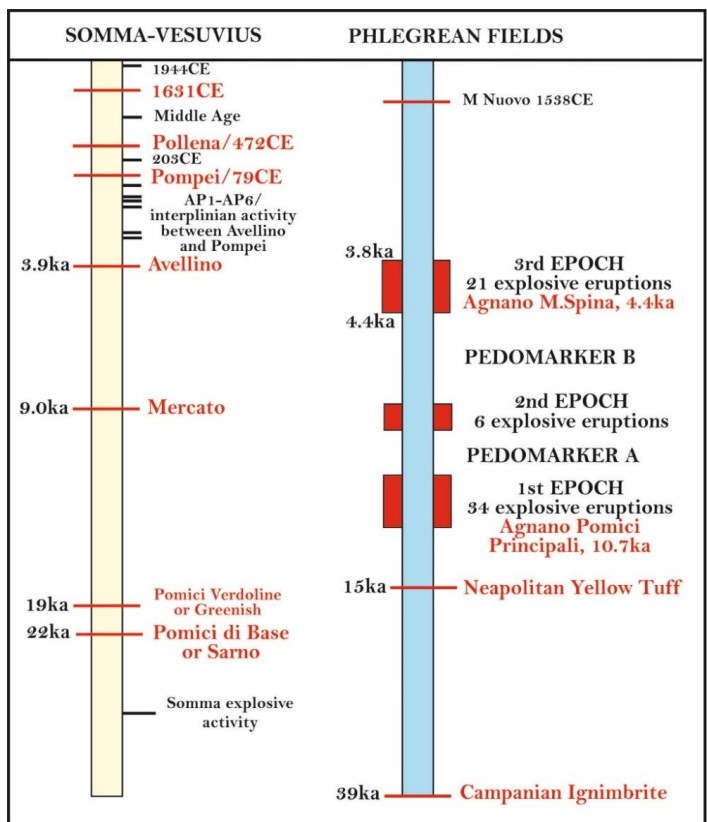

**Figure 2: Main explosive eruptions of Somma-Vesuvius and Phlegrean Fields volcanoes. The major explosive events are in red. The pedomarker A and pedomarker B refer to paleosol layers developed during eruptive quiescence.**

## 3. Materials and methods

The data and methods used in this study are summarized in Table 1 and Fig. 3 and further described in this section.

### 3.1. Dataset of field-based thickness measurements and related methods

A dataset of 6,671 field-based thickness measurements (Matano et al., 2023) have been collected during the field surveys and investigations for scientific and technical studies in the study area (Fig. 1) over the last decades.

The following methods have been applied for measuring thickness of the unconsolidated pyroclastic materials on the bedrock:

- Probing tests (PRBs): An iron rod (1.8 cm in diameter and up to 306 cm in length) was driven into the ground by hand or by a 0.03-kN hammer to measure depth of the underlying consolidated bedrock, indicating thickness of the fallout pyroclastic deposits as well. Each measurement represents the arithmetic mean of two or three measured values within a circle with a radius of 1-2 m to minimize the error associated with the local factors such as presence of cobbles, boulders, roots, colluvium and pumice layers.



- Penetration tests: Two types of penetration tests were implemented in this study. The Dynamic Cone Penetration tests (DPT–DL030) were performed by driving an iron rod with a cross-sectional area of 10 cm$^2$ into the ground by repeatedly raising a 0.3-kN weight for 20 cm and dropping it. It refers to the in-situ continuous measurement of rock/soil resistance to penetration up to 14 m depth which could also be an indirect measure for thickness of the fallout pyroclastic deposits when the probe fails to penetrate. The collected data by this method are in accordance with the probing test results and help interpret the stratigraphy as well. We also used the results of Standard Penetration Test (SPT), which is a common in-situ dynamic test for determining the geotechnical properties of subsurface soil such as relative density and shear strength parameters.

- Borehole (BH): The borehole stratigraphic data have been used for collecting thickness of fallout pyroclastic deposits.

- Hand-dug pits (HDPs): They were usually excavated manually (mainly 20×20×200 cm) near penetration test or geophysical survey sites for collecting further information on stratigraphy of the loose materials over the bedrock.

- Trenches (TRNs): They were excavated (1 m wide, 3 m long and 2-3 m deep) at the base of the slopes or along the intermediate morphological shelves using mechanical diggers for direct investigation of the pyroclastic deposit stratigraphy.

- Seismic surveys (SSs): Up to 10 m depth, the seismic reflection data of 3 bursts (direct, reverse and intermediate) were recorded by 20-24 geophones placed 3-5 m apart in a straight line on the ground surface. The seismic data revealed the geometry and stratigraphy of the ash layer along with the boundary between the consolidated bedrock and the overlaying loose materials.

- Outcrops (OCPs): Stratigraphy of several outcrops were analyzed across the study area for measuring thickness of the pyroclastic deposits.

To date, partial/total thickness measurements, method of investigation, the municipality territory to which the measurement points belong, and the geographic coordinates are recorded in a total of 6671 points (Matano et al., 2023). The spatial distribution of the measurement points is shown in Fig. 4. Matano et al. (2016) and Cuomo et al. (2021) have already used the measurements collected in Cervinara and Nocera Inferiore municipality territories for detailed thickness mapping with heuristic methods.

Earth System
Science
Data


**Figure 3: A flowchart showing how the dataset of field-based thickness measurements (Matano et al., 2023) is used to predict thickness of the fallout pyroclastic deposits. GPT: Geomorphological Pyroclastic Thickness; SAPT: Slope Angle Pyroclastic Thickness, SEPT: Slope Exponential Pyroclastic Thickness; STPW: stepwise regression; RF: random forest; SA: Simple Average; MV: Minimum Variance; OLS: Ordinary Least Squares; LAD: Least Absolute Deviation.**


**Figure 4: Spatial distribution of field-based thickness measurement points in each municipality: (a) Villa Santa Lucia; (b) Pozzilli; (c) Gioia Sannitica; (d) Nocera Inferiore; (e) Bagnoli Irpino; (f) Vitulano; and (g) Cervinara. The measurement points are subdivided into the training (n = 4294) and test (n = 1843) subsets.**






| Predictor variable | Description | Input data | Methodology | Tool |
|---|---|---|---|---|
| Altitude | It shows the elevation above sea level. | DEM[1] | N.A.[2] | N.A. |
| Aspect | It refers to the direction that the downhill slope faces. | DEM | Burrough and McDonell (1998); Ligas and Banasik (2011); Krakiwsky and Wells (1971); Lancaster and Salkauskas (1986); Hofmann-Wellenhof et al. (2001) | Aspect in ArcMap |
| Distance to hydrographic network | It indicates the distance to the hydrographic network. | ISPRA hydrographic network[3] | N.A.[1] | Euclidean Distance in ArcMap |
| Distance to source | It represents the distance to the eruptive vents. | Di Vito et al. (2008) | N.A. | Euclidean Distance in ArcMap |
| Flow accumulation | Flow accumulation for a cell refers to the number of cells that flow to it. | DEM | Jenson and Domingue (1988); Tarboton et al. (1991) | Flow Accumulation in ArcMap |
| Flow direction | The flow direction for a cell indicates the direction water will flow out of the cell. | DEM | Greenlee (1987); Qin et al. (2007); Tarboton et al. (1991) | Flow Direction in ArcMap |
| Initial thickness ($z_0$) of fallout pyroclastic deposits | It refers to the thickness of fallout pyroclastic deposits that erupted from the volcanos without the influence of denudational processes. | See Tables 2 and 3 | See section 3.1.2 | ArcMap |
| Curvature | It is the second derivative of the surface, or the slope-of-the-slope. | DEM | Moore et al. (1991); Zevenbergen and Thorne (1987) | Curvature in ArcMap |
| Modified Secondary Soil-Adjusted Vegetation Index ($MSAVI_2$) | It indicates healthy green vegetation. | Landsat 8 OLI | Qi et al. (1994) | Raster Calculator in ArcMap |
| Normalized Clay Index (NCI) | It is indicative of clay or hydroxyl-bearing minerals. | Landsat 8 OLI | Kienast-Brown et al. (2017) | Raster Calculator in ArcMap |
| Normalized Difference Vegetation Index (NDVI) | It shows healthy green vegetation. | Landsat 8 OLI[4] | Jensen (2015) | Raster Calculator in ArcMap |
| Plan curvature | It is in the direction of the maximum slope. | DEM | Moore et al. (1991); Zevenbergen and Thorne (1987) | Curvature in ArcMap |
| Profile curvature | It is perpendicular to the direction of the maximum slope. | DEM | Moore et al. (1991); Zevenbergen and Thorne (1987) | Curvature in ArcMap |
| Slope | It identifies the steepness of the ground surface. | DEM | Burrough and McDonell (1998); Ligas and Banasik (2011); Hofmann-Wellenhof et al. (2001) | Slope in ArcMap |



| Stream power index (SPI) | It characterizes the erosive power of flowing water. | DEM | Moore et al. (1991) | Raster Calculator in ArcMap |
|---|---|---|---|---|
| Stream transport index (STI) | It shows the erosive power of surface flow. | DEM | Moore and Burch (1986) | Raster Calculator in ArcMap |
| Topographic wetness index (TWI) | It is a proxy for soil moisture. | DEM | Beven and Kirkby (1979); Moore et al. (1991) | Raster Calculator in ArcMap |
| Topsoil Grain Size Index (TGSI) | It represents the fine sand content of the topsoil. | Landsat 8 OLI | Xiao et al. (2006) | Raster Calculator in ArcMap |

[1] Tarquini et al. (2007); [2] Not applicable; [3] https://geodati.gov.it/resource/id/ispra_rm:01ldro250N_DT (accessed on 8 Jan. 2024); [4] Operational Land Imager

**Table 1: The predictor variables for thickness modeling.**



### 3.2. Predictor variables

A list of the potential predictor variables for estimating thickness of fallout pyroclastic deposits is provided in Table 1. The value for each predictor variable is assigned to the measurement points based on a set of rasters in 30×30 m resolution.

### 3.2.1. Initial thickness ($z_0$) of fallout pyroclastic deposits

This predictor variable represents the overall thickness of fallout pyroclastic deposits emplaced by Late Quaternary explosive eruptions at a given location. In other words, it explains the thickness value that could be estimated at a location if erosional and/or depositional processes do not occur after the associated eruptive events. In fact, the current residual thickness of pyroclastic deposits that can be found at a certain location today is the result of the erosional and depositional processes that occurred after the eruptive events.

To obtain the initial thickness ($z_0$) of fallout pyroclastic deposits, the following approach is used: (1) collecting the isopach maps of the fallout deposits for the main volcanic eruptions (characterized by high explosivity index and great eruptive volume) in Campania region from literature (Tables 2 and 3); (2) georeferencing and digitizing each map; (3) applying an interpolation technique (i.e. Topo to Raster in ArcMap) to add intermediate isopaches in case of a significant gap between them; (4) assigning the average value of two isopaches of different thickness to the area between them, except

for the area enclosed by only one isopach; (4) combining all shapefiles into one; (4) computing $z_0$ of all volcanic eruptions for each feature in the shapefile; and (5) converting the obtained shapefile into a raster with 30×30m resolution and assigning the $z_0$ value to each field-based measurement point in the thickness dataset (section 3.1).

The isopach maps of the fallout deposits for the Somma-Vesuvius and Phlegrean Fields main eruptions are respectively listed in Table 2 and Table 3 with the related references. The Ischia tephra was not considered for $z_0$ calculation because of its insignificant thickness on the mainland. However, the old Roccamonfina tephra (>150 ka) has been almost entirely

eroded outside the volcanic edifice and we have considered the associated isopach map of pyroclastic deposits only in a semi-quantitative way, based on the results of Rouchon et al. (2008) and Giannetti et al. (2001).

### 3.2.2. Variables derived from DEM and satellite imageries

The predictor variables from DEM or satellite imageries (11 and 4 variables, respectively) are listed in Table 1 with a

definition, a brief description, and the methodology for obtaining them. Originally, the DEM is in 10×10m spatial resolution while the satellite imageries are in 30×30m spatial resolution. Raster resampling is, therefore, implemented after calculating the variables to obtain a resolution of 30×30m.

Other variables, as the distance to the hydrographic network and the source (i.e. eruptive vent) are also considered as predictor variables in this study. Further information is provided in Table 1.





| Eruption | Age | Reference |
|---|---|---|
| 1944 | 1944 C.E. | Cole and Scarpati (2010); Cubellis et al. (2016) |
| 1906 | 1906 C.E. | Arrighi et al. (2001); Cerbai and Principe, 1996 |
| 1822 | 1822 C.E. | |
| 1730 | 1730 C.E. | |
| 1723 | 1723 C.E. | |
| 1707 | 1707 C.E. | |
| 1682 | 1682 C.E. | |
| 1631 | 1631 C.E. | Rolandi et al. (1993a); Rosi et al. (1993); Bertagnini et al. (2006) |
| Third medieval eruption | 1140±60 yr BP | Rolandi et al. (1998) |
| Second medieval eruption | 1290±40 to 1440±60 yr BP | |
| First medieval eruption | | |
| Pollena | 472 C.E. | Rolandi et al. (2004); Sulpizio et al. (2007, 2005); Rosi and Santacroce (1983); Delibrias et al. (1979) |
| Pompei | 79 C.E. | Sigurdsson et al. (1985); Luongo et al. (2003a, 2003b); Zanella et al. (2012) |
| AP6 | 203 B.C.E. | Andronico and Cioni (2002); Rolandi et al. (1998); Somma et al. (2001) |
| AP5 | | |
| AP4 | | |
| AP3 | 2710±60 yr BP | |
| AP2 | 3000±200 yr BP; 3225 to 1140 yr BP | |
| AP1 | 3220±65 yr BP 3420±100 yr BP | |
| Avellino | 3.9 ka BP | Rolandi et al. (1993b); Di Vito et al. (2019); Sevink et al. (2011); Sulpizio et al. (2010a, 2010b) |
| Mercato | 9 ka BP | Rolandi et al. (1993a); Aulinas et al. (2008) |
| Pomici Verdoline | 19 ka BP | Cioni et al. (2003); Arnò et al. (1987) |
| Pomici di Base | 22 ka BP | Bertagnini et al. (1998); Andronico et al. (1995); Arnò et al. (1987) |
| Codola | 25.10±0.40 ka BP 30.02±0.42 cal ka BP 33 ka BP inferred calendar age | Di Vito et al. (2008) |

**Table 2: The Somma-Vesuvius eruptions used for elaborating isopach maps.**

| Eruption | Age | Reference |
|---|---|---|
| Averno 2 | 3.8 ka BP | Costa et al. (2009) |
| Astroni | 3.8 ka BP | Costa et al. (2009); Di Vito et al. (2021) |
| Agnano Monte Spina | 4.4 ka BP | De Vita et al. (1999); Costa et al. (2009) |
| PaleoAstroni 2 | 4.7 ka BP | Di Vito et al. (2021) |
| Agnano 3 | 5.2 ka BP | |
| Agnano Pomici Principali | 12.5 ka BP | Orsi et al. (2004); Di Vito et al. (1999) |
| Neapolitan Yellow Tuff | 15 ka BP | Scarpati et al. (1993) |
| TAU1-e | 22 to 23 cal ka BP | Di Vito et al. (2008) |
| Masseria del Monte | 29 ka BP | Albert et al. (2019) |
| SMP1-d | 33 to 36 cal ka B.P. | Di Vito et al. (2008) |
| Taurano | 33 to 36 cal ka B.P. | |
| Campanian Ignimbrite | 39 ka BP | Costa et al. (2012); Giaccio et al. (2008); Cappelletti et al. (2003) |
| Santa Lucia | 47.5±2.6 cal ka BP 50.95±2.98 ka BP | Di Vito et al. (2008) |
| CA1-a | 55 to 105 ka BP | Di Vito et al. (2008) |

**Table 3: The Phlegrean Fields eruptions considered for elaborating isopach maps.**



### 3.3. Methods for thickness modeling

#### 3.3.1. Previous studies

To date, four approaches have been proposed for modeling thickness ($z$) of the fallout pyroclastic deposits. The Slope Angle Pyroclastic Thickness (SAPT) model estimates $z$ by linking the initial thickness ($z_0$) of fallout pyroclastic deposits erupted from the volcanos with the slope angle (De Vita et al., 2006; De Vita and Nappi, 2013). In this model, some thresholds for slope angle were derived by field measurements in Mt. Sarno and Mt. Lattari. The Geomorphological Index

Soil Thickness (GIST) model is an empirical model that combines morphometric, geomorphological and geological features (Catani et al., 2010) for estimating soil thickness in areas where bedrock weathering is the main soil forming process (Mercogliano et al., 2013; Segoni et al., 2013), but applied to the areas covered by the fallout pyroclastic deposits as well (Rossi et al., 2013). In this article, the GIST model was not implemented because the fallout pyroclastic deposits are of allochthonous origin and bedrock lithology does not control their thickness (De Vita et al., 2006; Del Soldato et

al., 2018). Del Soldato et al. (2016) proposed the Geomorphological Pyroclastic Thickness (GPT) model as a combination of the SAPT and GIST models. Comparing performance of GIST, SAPT and GPT models indicated that $z$ is mainly controlled by $z_0$ and slope gradient. Therefore, the Slope Exponential Pyroclastic Thickness (SEPT) model was developed based on these two parameters (Del Soldato et al., 2018).

#### 3.3.2. Proposed methods: Random Forest

For spatial modeling, a wide range of machine learning techniques are available including logistic regression analysis, random forest (RF), support vector machine and artificial neural networks. Among these techniques, RF showed the best performance for classification and prediction. It is a shallow ensemble learning algorithm that could be tuned with few parameters (Liu et al., 2023 and references therein). The principles of decision trees and bagging are implemented for building random forests. Bagging applies bootstrap sampling of the training data for building decision trees and aggregates

the predictions across all the trees which reduces the overall variance and improves the predictive performance. The RF uses a random subset of variables at each split while growing a decision tree during the bagging process to generate a more diverse set of trees which helps lessen tree correlation beyond bagged trees and noticeably increase the predictive power.

   After splitting a given dataset randomly into training and test subsets (Fig. 4), the RF regression modeling could be applied

as follows: (1) generating a RF model using the training subset; (2) calculating the variable importance for the established model; (3) applying the constructed RF model to the test subset and evaluating the results; and (4) implementing the trained RF model for making predictions in the unknown locations.

   The R packages "rsample" (Frick et al., 2022) and "ranger" (Wright and Ziegler, 2017) are used for data splitting and modeling, respectively. We considered 70% of the whole dataset as training subset and the rest as test subset in data

splitting (Fig. 4). For training a model, different values are assigned to each hyperparameter, including the number of variables to possibly split at each node ($m_{try}$), minimal node size to split at (*min.node.size*), sample with/without replacement (*replace*) and fraction of observations to sample (*sample.fraction*). A data frame from all possible combinations of $m_{try}$, *min.node.size*, *replace* and *sample.fraction* was then generated, the RF model was trained for each combination and the best one was selected regarding root mean square error (RMSE) and mean absolute error (MAE)

(see section 3.2.3 for more information). The optimum number of trees (*num.trees*) was finally investigated by running the RF model for 50 different *num.trees* values between 0 and 1000. Using the determined hyperparameters, the RF model is trained for making predictions, performance of the model is evaluated and the importance of variables is calculated by:



(1) the Gini index (Fig. 10b) which indicates the number of times a variable is responsible for a split and the impact of that split divided by the number of trees; and (2) the permutation importance (Fig. 10c) which calculates prediction

accuracy in the out-of-bag observations and recomputes the prediction accuracy after eliminating any association between the variable of interest and the outcome by permuting the values of the variable under evaluation. The difference between the two accuracy values is the permutation importance for the given variable from a single tree. The average of importance values for all trees in a RF then gives the RF permutation importance of this variable.

### 3.3.3. Proposed methods: Stepwise regression model


Multiple linear regression (MLR) is used to analyze the relationship between a single response variable (dependent variable) and two or more independent variables (predictor variables). Assuming we store $P$ predictor variables (p=1,…,P) for N locations (i=1,…,N) in a matrix $X$ $\{x_{i,p}\}$, we could simply predict thickness $z$ $\{z_i\}$ using multiple linear regression:

$$z = \beta X + \varepsilon \tag{1}$$

with $\beta = [\beta_1, … , \beta_P]'$ the vector of regression coefficients and $\varepsilon$ a vector of i.i.d. error terms. However, all the $P$ variables are not necessarily relevant for making prediction and more accurate predictions may be obtained by a subset $\tilde{X}\{\tilde{x}_{i,p}, p = 1, … , \tilde{P}; \tilde{P} < P\}$ of predictor variables. Then, the final model can be written as follows:

$$z = \delta \tilde{X} + \eta \tag{2}$$

with $\delta = [\delta_1, … , \delta_{\tilde{P}}]'$ the vector of the selected best $\tilde{P} < P$ variables and $\eta$ the new vector of error terms.

Different methods such as forward selection, backward elimination and stepwise regression (STPW) are used to this aim. All these methods are based on a series of automated steps (Taylor and Tibishirani, 2015). A forward-selection approach initially assumes no predictor variable and adds the most statistically significant variable, one by one, until no more variable remains. On the contrary, the backward elimination approach initially includes all predictor variables and then eliminates the least statistically significant variables one by one. However, the STPW method is a combination of forward

selection and backward elimination. As with forward selection, the procedure starts with no variables and adds variables using a pre-specified criterion. At every step, the procedure also considers the statistical consequences of dropping the previously included variables. The STPW method is applied in this article with the R package "StepReg" (Li et al., 2020).

### 3.3.4. Proposed methods: Combination approaches

The RF, STPW, GPT, SAPT and SEPT models possess their own strengths and weaknesses. Previous studies showed

that a combination of the predictions obtained with different methods allows for more accurate estimations (e.g., in the case of time series, see Eliott and Timmermann, 2004; Chan and Pauwels, 2016). Therefore, combination-based predictions are commonly used in many applicative fields (e.g., Cui et al., 2021; Nti et al., 2020; Wong et al., 2007; Yang, 2018).

One of the most common approaches for combining predictions is the stacking ensemble (Ganaie et al., 2022). It trains

different models on the same dataset and generates predictions that become the input of a superior model (known as a second-level model; see Ribeiro, 2020). The fundamental concept behind stacking is that the optimal combination of the predictions of different models achieves better predictive performance compared to those obtained with single models. Let us define, for each $i$-th location, a vector consisting of $K$ $(k = 1, … , K)$ alternative predictive models $\hat{z}_i = [\hat{z}_{i,1}, \hat{z}_{i,2}, … , \hat{z}_{i,k}, … , \hat{z}_{i,K}]'$ which can be obtained considering some inputs as shown in Fig. 4. Then, a final prediction at

the $i$-th location can be defined according to:

$$\tilde{z}_i = f(\hat{z}_i, \omega) \tag{3}$$





with $\boldsymbol{\omega} = [\omega_1, \omega_2, \dots, \omega_k, \dots, \omega_K]'$ be the vector of $K$ weights associated with the $K$ different competing spatial predictive models. In other words, the stacking ensemble $\tilde{z}_i$ is a function of the $K$ predictions with different base models for the same location $\hat{\mathbf{z}}_i$. In particular, we assume a linear function:

$$\tilde{z}_i = \boldsymbol{\omega}'\hat{\mathbf{z}}_i = \sum_{k=1}^{K} \omega_k \hat{z}_{i,k} \qquad (4)$$


In this framework, an important issue is the selection of the combination weights $\boldsymbol{\omega}$. To this aim, we can use subjective or objective weighting systems based on either expert evaluations or some statistical criteria. In this paper, we compare the performance of four objective weighting systems and choose the best one.

As the first approach, we consider the simple average (SA) combination in which the $K$ competing models are weighted equally, i.e. $\boldsymbol{\omega}_{SA} = [\frac{1}{K}, \dots, \frac{1}{K}]'$. Despite its simplicity, this approach empirically provides a better performance compared to more sophisticated alternatives (Hsiao and Wan, 2014). Another commonly adopted approach for optimal selection of combination weights is based on variance minimization criterion (MV, see Bates and Granger, 1969; Newbold and Granger, 1974). Given a set of $K$ competing predictive models, the weights are chosen by minimizing the variance of the prediction errors:


$$\min_{\boldsymbol{\omega}} \boldsymbol{\omega}'\boldsymbol{\Sigma_e}\boldsymbol{\omega}, \qquad with \ \boldsymbol{\iota}'\boldsymbol{\omega} = 1 \qquad (5)$$


with $\boldsymbol{\iota} = [1,1,\dots,1]'$ a vector of ones and $\boldsymbol{\Sigma_e}$ the $K \times K$ covariance matrix associated to the prediction errors of the $K$ competing models. The optimal solution to this minimization problem is given by:

$$\boldsymbol{\omega}_{MV} = \frac{\boldsymbol{\Sigma_e^{-1}}\boldsymbol{\iota}}{\boldsymbol{\iota}'\boldsymbol{\Sigma_e^{-1}}\boldsymbol{\iota}} \qquad (6)$$

where $\boldsymbol{\Sigma_e^{-1}}$ is the inverse of the covariance matrix, also known as the precision matrix.

The third approach implemented for choosing combination weights $\boldsymbol{\omega}$ is based on Ordinary Least Squares (OLS) regression (Granger and Ramanathan, 1984), where $\boldsymbol{\omega}$ can be chosen by considering the following linear regression:


$$z_i = \omega_0 + \sum_{k=1}^{K} \omega_k \hat{z}_{i,k} + \varepsilon_i \qquad (7)$$

with $\omega_0$ the constant term, $\omega_k$ the generic k-th weight associated with the k-th competing model and $\varepsilon_i$ an i.i.d. error term. According to OLS combination, the weight vector $\boldsymbol{\omega} = [\omega_1, \dots, \omega_K]'$ is obtained by solving the following minimization problem:


$$\min_{\omega_1,\dots,\omega_K} \sum_{i=1}^{N} \left( z_i - \sum_{k=1}^{K} \omega_k \hat{z}_{i,k} \right)^2 \qquad (8)$$

The OLS approach requires computing the weights in a training subset and using the selected ones in a test subset. It has the advantage of generating unbiased combined predictions without the need to investigate the bias for the individual models. This weighting approach is, however, sensitive to outliers. To address this issue, previous studies proposed the Least Absolute Deviation (LAD) combination approach, based on the minimization of the absolute loss function (Nowotarski et al., 2014):


$$\min_{\omega_1,\dots,\omega_K} \sum_{i=1}^{N} \left| z_i - \sum_{k=1}^{K} \omega_k \hat{z}_{i,k} \right| \qquad (9)$$





### 3.3.4. Accuracy evaluation

To evaluate and compare performance of the $K$ predictive models and their combinations, Root Mean Square Error (RMSE) and the Mean Absolute Error (MAE) are computed. Let us first define the prediction error $e_{i,k}$ of the $k$-th model (including combinations of the models) for the observed $i$-th location:

$$e_{i,k} = z_i - \hat{z}_{i,k}.$$

The $RMSE_k$ and $MAE_k$ are defined as:

$$RMSE_k = \sqrt{\frac{1}{N}\sum_{i=1}^{N} e_{i,k}^2} = \sqrt{\frac{1}{N}\sum_{i=1}^{N}\left(z_i - \hat{z}_{i,k}\right)^2} \tag{10}$$

$$MAE_k = \sqrt{\frac{1}{N}\sum_{i=1}^{N} |e_{i,k}|} = \sqrt{\frac{1}{N}\sum_{i=1}^{N} |z_i - \hat{z}_{i,k}|} \tag{11}$$

Notice that MAE loss is less affected by outliers than RMSE and we prefer the models with lower MAE in case of ambiguity.

Moreover, we consider the Equal Predictive Accuracy test (EPA; Diebold and Mariano, 2002) for investigating the statistical difference between the $K$ competing models and their combinations. Given two competing models $k$ and $k'$, we define a generic loss function $g(\cdot)$ of the prediction errors, $g(e_{i,k})$ and $g(e_{i,k'})$. In our case, we consider squared and absolute losses (RMSE and MAE, respectively). Let us define the loss differential vector $\boldsymbol{d} = [d_1, d_2, \dots, d_N]'$, where for each generic $i$-th location:

$$d_i = g(e_{i,k}) - g(e_{i,k'}) \tag{12}$$

Under the null hypothesis, the vector $\boldsymbol{d}$ has zero mean and the two competing models $k$ and $k'$ have the same predictive accuracy. Under the alternative hypothesis, the two models are statistically different, and the best model is the one associated with the lowest statistical loss. In practice, the EPA test can be simply applied by regressing the loss differential vector $\boldsymbol{d}$ with a constant vector $\boldsymbol{\iota}$ of ones and by conducting inference with robust standard errors to account for possible heteroskedasticity.

### 4. Description of the field-based thickness dataset and related predictor variables

We first explain the process of creating a subset from the dataset of Matano et al. (2023) (n = 6671) to achieve our research objectives. Briefly, 41 measurements belonging to Forio and Procida municipalities are excluded because they are not located on the mainland. The subset is populated with 18 predictor variables and visualized in Figs 5-8 to have an idea of the available data for detailed elaboration. It is noteworthy that the "partial" thickness measurements (n = 493) are then culled and the remaining 6137 points (Fig. 2) are considered for thickness modelling in the following section. The field-based thickness measurements range between 0 and 1450 cm. Most of the measurements refer to "total" thickness, representing thickness of fallout pyroclastic deposits from ground surface to the underlying bedrock (Fig. 5). The median of "partial" thickness values is three times greater than that of "total" thickness (approximately 200 and 60 cm, respectively). This difference is because the "partial" thickness values are carried out in areas normally characterized by a thickness greater than the maximum survey depth that the measurement method allows, which for the most widely used method (the probing tests and the hand-dug pits) is respectively about 300 cm and 200 cm. The average and range of thickness values mainly show the limitations of the measurement methods. Probing test is the leading methodology



implemented in field surveys and the measured values usually range from 10 to 300 cm. The recorded values are often

<10 cm in outcrops and 200-800 cm in the measurements through SPT and boreholes. The surveys were mainly conducted in Cervinara, Nocera Inferiore and Vitulano territories (43, 24 and 24%, respectively; Fig. 5). The median thickness values are above 60 cm in Nocera Inferiore and Cervinara while they fluctuate around 35 cm in the other municipalities. The values of predictor variables assigned to the "total" and "partial" measurement points are almost similar, but the minor differences have some interesting interpretations (Fig. 6). Compared to the stations for "total" measurements, the median

values in Fig. 6 show that thickness is partially recorded in the measurement points with lower altitude, distance to source and slope degree (Fig. 6a, e and m). In addition, the distance of these stations to the hydrographic network is greater than that of "total" thickness measurement points (Fig. 6d). This last aspect confirms that the bedrock is usually not reached by investigation, and therefore the thickness is greater, as one moves away from the hydrographic network, the torrential erosive intensity being lower. The similarity between Fig. 5d (i.e. the three boxplots on the left) and 6r probably reveals

that $z_0$ is a good indicator of z.

The measurement points are also categorized regarding the measurement methods (Fig. 7) to underline the category-based variation of the predictor variables. The insignificant variation of the variables for the stations investigated through borehole stratigraphic data explains the few field-based observations under this category. Standard Penetration Tests and seismic surveys are the preferred methods in the low-altitude locations far from the hydrographic network and near the

source (Fig. 7a, d and e). On the contrary, thickness of the fallout pyroclastic deposits is investigated via trenches and outcrops in the measurement points farthest from the source (Fig. 7e). The lowest computed $z_0$ values are related to these stations as well (Fig. 7r). Likewise Fig. 7, range of the predictor values are compared in different municipalities (Fig. 8). It provides some spatial information on each municipality and the measurement points belonging to them. For instance, Bagnoli Irpino and Villa Santa Lucia occur in the highest and lowest altitudes, respectively (Fig. 8a). The latter has the

lowest distance to the hydrographic network along with the minimum calculated $z_0$ and measured z values (Fig. 8d, 8r and 5d, respectively). Cervinara and Nocera Inferiore are the closest municipalities to the source (Fig. 8e), where the greatest $z_0$ values are calculated (Fig. 8r), above 65% of the measurements are performed (Fig. 5c), and the highest z values are recorded (the right panel of Fig. 5d). The discussion in this section indicates a relatively high level of heterogeneity of the variables. The presence of many data outliers can also be visually verified in Figs. 5-8. These

characteristics of the data suggest a complex relationship between the predictor variables and thickness of fallout pyroclastic deposits, which can be better modeled by means of machine learning techniques rather than the standard models employed in the literature.

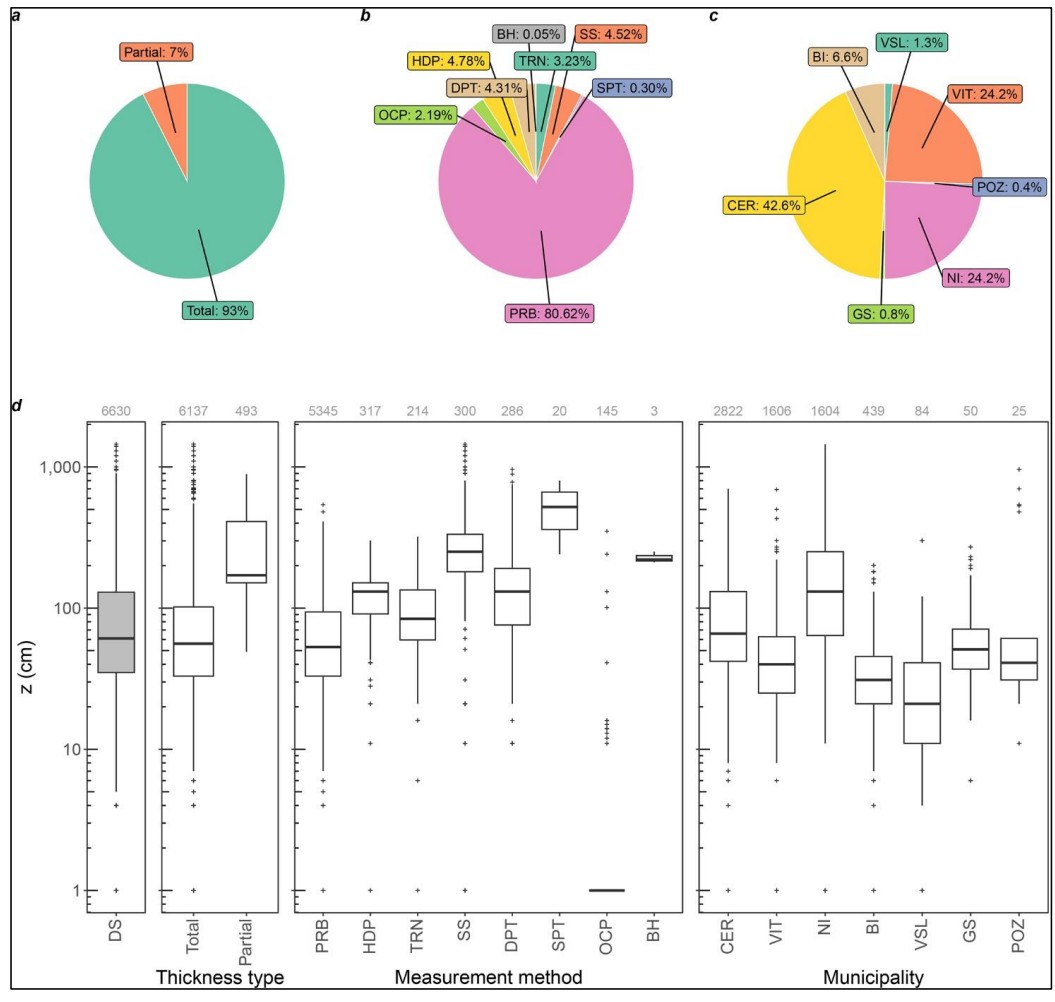

**Figure 5: An overview of the field-based thickness measurements of the fallout pyroclastic deposit in study area: (a) proportion of total and partial measurements; (b) proportion of the measurements regarding the methodology; (c) proportion of the measurements in each municipality; and (d) variation of thickness considering the whole dataset, total/partial measurements, measurement method and municipality. DS: dataset; OCP: outcrop; PRB: probing test; TRN: trench; HDP: hand-dug pit; DPT: Dynamic Cone Penetration tests; BH: borehole; SS: seismic survey; SPT: Standard Penetration Test; NI: Nocera Inferiore; BI: Bagnoli Irpino; CER: Cervinara; VIT: Vitulano; GS: Gioia Sannitica; POZ: Pozzilli; and VSL: Villa Santa Lucia. It is noteworthy that the "partial" measurements are excluded for modelling thickness in this study.**



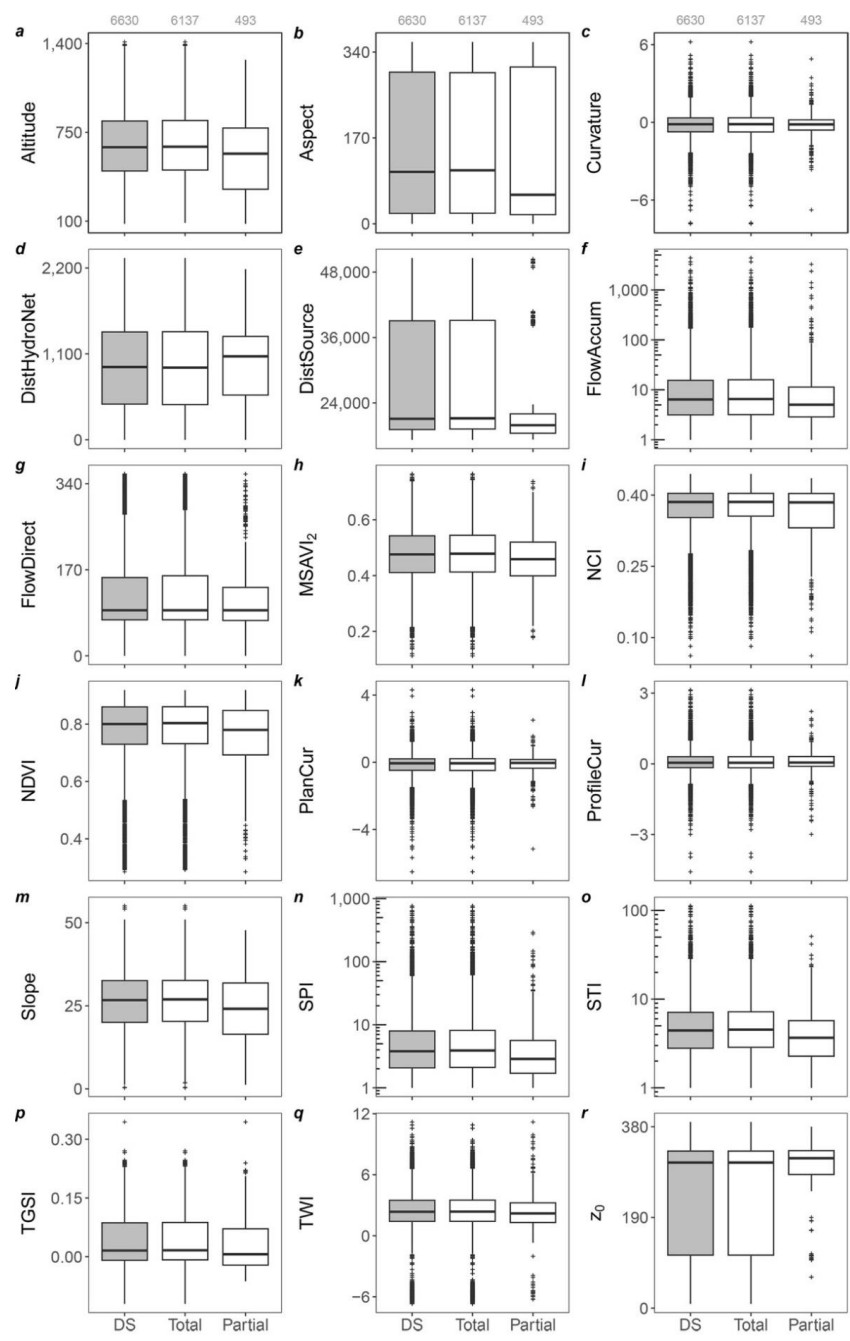

**Figure 6: Range of the values for predictor variables referred to the whole dataset (DS) and to the total/partial measurements. DistHydroNet: Distance to hydrographic network; DistSource: Distance to source; FlowAccum: Flow accumulation; FlowDirect: Flow direction; PlanCur: Plan curvature; ProfileCur: Profile curvature. It is noteworthy that the "partial" measurements are excluded before thickness modelling in this study.**




**Figure 7: Range of the values for the predictor variables referred to the whole dataset (DS) and the different measurement methods (from PRB to BH). See caption of Fig. 5 for the abbreviations. It is noteworthy that the "partial" measurements (n = 493) are excluded before thickness modelling in this study.**





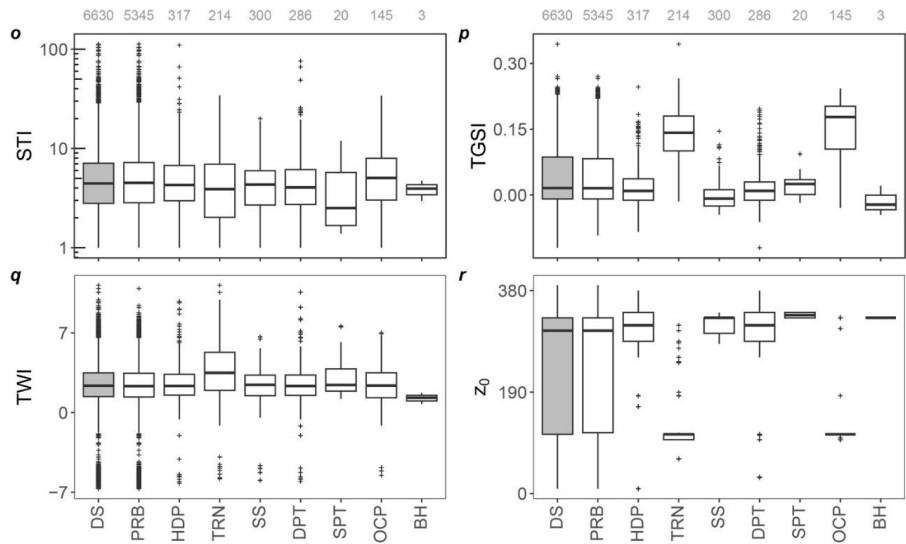


**Figure 7 (continued)**







**Figure 8: Range of values for the predictor variables referred to the whole dataset (DS) and the municipalities (from CER to POZ). See caption of Fig. 5 for abbreviations. It is noteworthy that the "partial" measurements (n = 493) are excluded before thickness modelling in this study.**




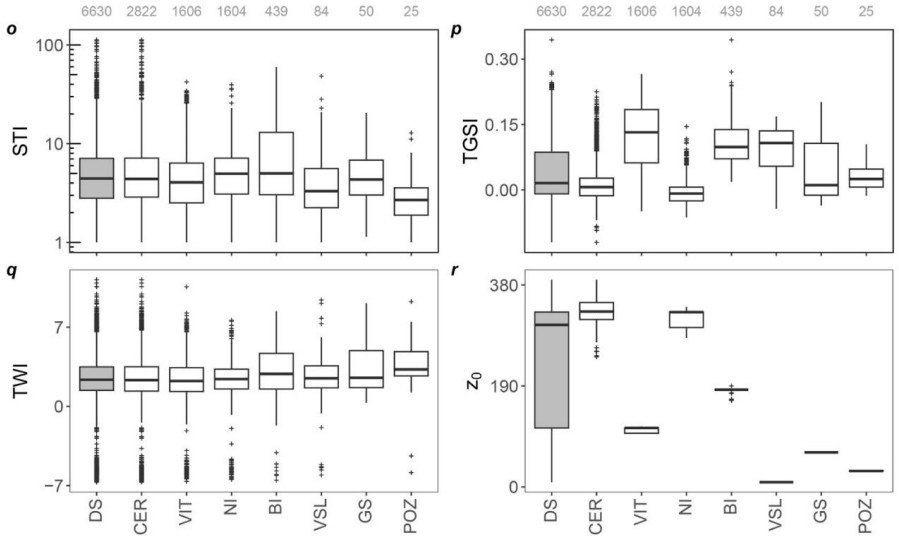

**Figure 8 (continued)**

## 5. The estimated thickness of fallout pyroclastic deposits: Results

In this article, we proposed applying RF and STPW approaches that train a model on the training subset and make estimations on the test subset. The dataset of 6137 measurement points was, therefore, randomly divided into training (n = 4294) and test (n = 1843) subsets (Fig. 4) and the same subsets were used in all predictive models to enable evaluating performance of the competing models. On the training data, the thickness is estimated by GPT, SAPT, SEPT, STPW and RF models. The predictions for different models are, then, combined according to the methods in Section 3 (Fig. 3).

Finally, the prediction errors are computed on both train and test subset and the predictive accuracy tests are implemented for evaluating differences in the test subset. This section explains the results in detail.

### 5.1. Training stepwise regression and Random Forest

The STPW model is used for selecting the best subset of variables in terms of explicative power for the pyroclastic thickness deposit. Given an initial set of 18 independent variables, only 8 relevant ones are chosen by the STPW model

for making predictions (Table 4): distance to the hydrographic network, distance to source, altitude, $z_0$, aspect, plan curvature, $MSAVI_2$ and NCI (for a detailed description of the variables see Table 1). All these variables are very relevant in determining thickness of fallout pyroclastic deposits as they control the erosion-deposition processes. The estimated parameters shown in Table 4 are then used for predicting the thickness values in the test subset.

In order to train Random Forest model representatively, different values are assigned to $m_{try}$, *min.node.size*, *replace* and

*sample.fraction* and a list of all possible combinations of the hyperparameters (882 in our case) is generated. Random forest is then trained for all combinations and the optimum value for each hyperparameter ($m_{try} = 5$; *min.node.size* = 17; *replace* = True; and *sample.fraction* = 0.632) is determined based on the model with the least error. The out-of-bag error is then investigated for different number of trees and *num.trees* = 530 is determined for training a model with the least error (Fig. 10a). Fig. 10b and 10c shows that altitude, $z_0$, NDVI, distance to hydrographic network, NCI and TGSI account

for the most important variables in training the model based on both variable importance metrics (i.e. impurity and



permutation). Compared to the STPW model, distance to source is the only variable excluded before RF modelling to avoid unrealistic estimations.

|  | Estimated parameter | Standard error | p-value |
|---|---|---|---|
| Intercept | 2.02E+08 | 2.16E+07 | 1.02E-14 |
| DistSource | 9.25E+00 | 3.45E+02 | 9.79E-05 |
| Altitude | -8.28E+04 | 7.66E+03 | 6.53E-21 |
| DistHydNet | 2.19E+04 | 2.79E+03 | 5.63E-09 |
| $z_0$ | 2.87E+05 | 3.43E+04 | 7.81E-11 |
| $MSAVI_2$ | -1.02E+06 | 3.06E+07 | 9.73E-05 |
| PlanCur | 2.65E+06 | 2.55E+06 | 2.97E-05 |
| Aspect | 1.63E+04 | 1.25E+04 | 1.90E-05 |
| NCI | -2.51E+08 | 8.09E+07 | 1.92E-03 |

**Table 4: List of the variables selected by STPW model for making predictions in the training subset. The estimated parameters, standard errors and p-values are also reported.**

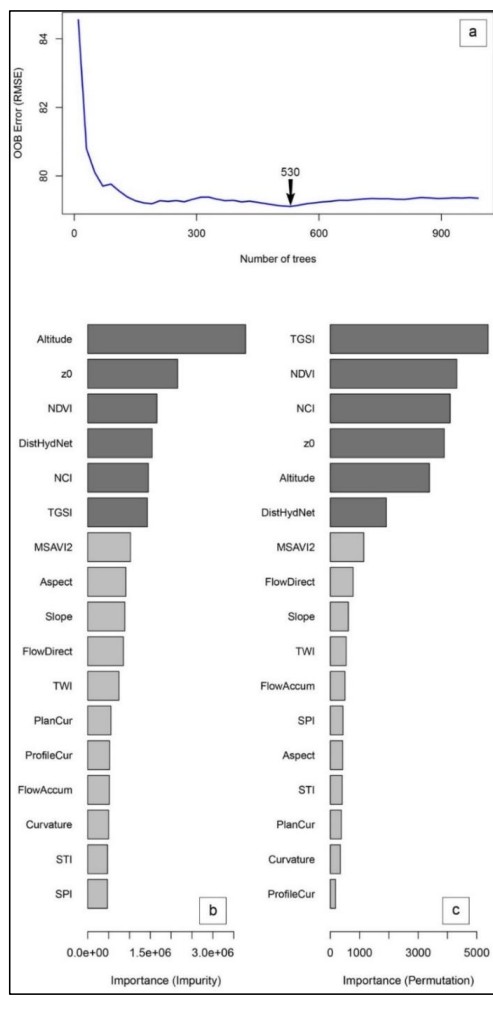


**Figure 10: (a) Variation of the out of bag error against number of trees; (b) Variable importance based on the impurity; and (c) variable importance based on permutation.**



### 5.2. Predicted thickness values

Considering the prediction accuracy of the single models applied to the training subset (Table 5), both STPW and RF
models improve the predictive accuracy measures compared with the literature approaches (i.e., GPT, SAPT and SEPT),
suggesting that the predictor variables discussed in section 5.1 play a crucial role in estimating thickness of fallout
pyroclastic deposits. However, the RF model (RMSE = 79.11 and MAE = 46.44) outperforms all the other approaches
(RMSE > 89 and MAE > 55).

| Category | Model | Train | | Test | |
|---|---|---|---|---|---|
| | | RMSE | MAE | RMSE | MAE |
| Single model | GPT | 95.36 | 56.91 | 94.21 | 56.93 |
| | SAPT | 184.52 | 157.10 | 107.45 | 58.05 |
| | SEPT | 107.20 | 59.75 | 187.95 | 160.31 |
| | STPW | 89.60 | 55.25 | 92.35 | 55.20 |
| | RF | 79.11 | 46.44 | 82.46 | 48.36 |
| Combination approach | SA | 91.64 | 60.82 | 94.22 | 61.86 |
| | MV | **79.05** | 46.05 | 82.51 | 47.97 |
| | OLS | 79.11 | 46.38 | **82.42** | 48.27 |
| | LAD | 80.83 | **44.03** | 83.22 | **45.12** |

**Table 5: Prediction accuracy results. Best model in bold. GPT: Geomorphological Pyroclastic Thickness; SAPT: Slope Angle Pyroclastic Thickness; SEPT: Slope Exponential Pyroclastic Thickness; STPW: Stepwise regression; RF: Random Forest; SA: Simple Average; MV: Minimum Variance; OLS: Ordinary Least Squares; LAD: Least Absolute Deviation.**

In the next step, the thickness predictions for the training subset (obtained from the single models) underwent a
combination approach using the weighting schemes in Table 6. Except for the SA method that assigns equal weights to
the models, all the weighting schemes assign the largest weight to RF, that is the most accurate one among the single
models. The accuracy measures for the combination models are shown under the "combination approach" category in
Table 5. According to the results for the training subset (Table 5), the RMSE shows that the predictions of the SA approach
are the worst, while the MV approach provides better accuracy than the RF model. The MAE function indicates that the
LAD method improves the accuracy about 5% compared to the RF model.

| Weighting scheme | GPT model | SAPT model | SEPT model | STPW model | RF model |
|---|---|---|---|---|---|
| SA | 0.2 | 0.2 | 0.2 | 0.2 | 0.2 |
| MV | 0.04 | -0.03 | -0.07 | 0.07 | 0.99 |
| OLS | 0.00 | 0.00 | 0.01 | 0.00 | 0.99 |
| LAD | -0.1 | 0.03 | -0.08 | -0.09 | 0.96 |

**Table 6: The weights for making predictions in the training subset with combination approaches. SA: Simple Average; MV: Minimum Variance; OLS: Ordinary Least Squares; LAD: Least Absolute Deviation**

Then, both single models and the alternative combination approaches for the test subset are compared in terms of RMSE
and MAE (see the last two columns of Table 5). Among the single models, the RF provides the most accurate results
(RMSE = 82.46 and MAE = 55.20), but combination of the RF predictions with those of the other models enhances the





accuracy. In particular, the OLS approach reduces RMSE to 82.42 and the LAD method lowers MAE to 45.12. Regarding
less sensitivity of MAE to data outliers, the LAD combination could be considered the most representative model. The
LAD combination improved the accuracy by 7.2% compared to the RF model. The improvement is above 26% respect to
the GPT model, the best single model proposed in the previous studies.

Finally, we investigate whether the differences in predictive performance are statistically significant. Table 7 shows the
pairwise comparison of the RF model as the best single model with all the combination approaches. A negative constant
value indicates that the RF model has a lower average prediction error than the combination approach, while a positive
value suggests that the predictions of the combination approach are more accurate. In terms of squared error loss, the OLS
combination approach provides the most accurate predictions than the RF model, but it is noteworthy that RMSE and
OLS are both sensitive to data outliers as explained in "3.2.2. Proposed methods". Regarding the MAE, statistically
significant improvement is observed when combination approaches are applied (except for the SA method; $p < 0.01$). The
greatest statistically significant constant value of the LAD method demonstrates that this combination technique is suitable
for predicting thickness of fallout pyroclastic deposits in unmeasured locations. The results are in accordance with those
in Table 5.

| Accuracy measure | | RF vs. SA | RF vs. MV | RF vs. LAD | RF vs. OLS |
|---|---|---|---|---|---|
| RMSE | Constant | -2076.229*** (432.758) | -7.006 (10.328) | -125.269 (126.711) | 5.971 (6.188) |
| MAE | Constant | -13.493*** (0.953) | 0.390*** (0.066) | 3.245*** (0.397) | 0.091*** (0.013) |

***$p < 0.01$

**Table 7: Equal predictive accuracy tests applied to the test subset (n = 1843). The heteroskedasticity robust standard errors**
**are in parentheses.**

### 6. Discussion

In this section, cumulative probability of the estimated thickness (for both train and test subsets) by various methods are
compared in Fig. 11a,b. Regarding the single models (Fig. 11a), significant underestimation of the SEPT model and
noticeable overestimation of the SAPT model are evident. These models have the highest RMSE and MAE values among
the single models (Table 5). Distribution of the values obtained by the GPT, RF and STPW models share more similarity
with that of field-based measurements. Although the STPW model performs better than RF in predicting the smaller
values, the RF model outperforms the single models which agrees with Table 5. Taking Fig. 11b into account, all
combination techniques work more effectively than the LAD approach in the upper 25% of thickness values. The overall
estimations of LAD approach are, however, more consistent with the field-based measurements, being confirmed by the
accuracy measures in Table 5.

The results of RF model and LAD approach are also visualized in Fig. 11c for the sake of comparison, representing that
the values estimated by former are greater than those of the latter. The LAD approach outperforms the RF model in a
wide range of the field-based thickness values, and it is, therefore, the best model for predicting thickness of fallout
pyroclastic deposits in the study area. Fig. 11d demonstrates that the LAD estimations are less biased.

The estimated thickness values of fallout pyroclastic deposits by the RF model and the LAD combination approach in
Vitulano, Cervinara and Nocera Inferiore are visualized in Figs. 12-14 to investigate the differences between the spatial
patterns. Although the spatial distribution remained generally unchanged, legends of the maps reveal that the estimated
thickness values by the LAD combination approach decreased as shown before (Fig. 11c). For instance, in Vitulano (Fig.





12), the estimations range from 10-196 cm in the RF model which is reduced to 1-177 cm in the LAD combination approach. This is also evident in the most estimations of the bottom-right box in which the values reduce from 66-94 cm to about 32-66 cm or in the top-left corner of the map: A (47-196 cm) > C (32-66 cm) > B (32-47cm) in the RF model vs. A (47-94 cm) > C (32-47 cm) > B (1-32 cm) in the LAD combination approach. The same decline could be observed in Cervinara and Nocera Inferiore as well. To facilitate a quick comparison, two boxes are drawn on Figs. 13 and 14 that highlight the sectors with a clear change. A few estimations exceed 113 cm in the lower panel of Fig. 13, being contrary to the upper panel generated by the RF model in Cervinara.

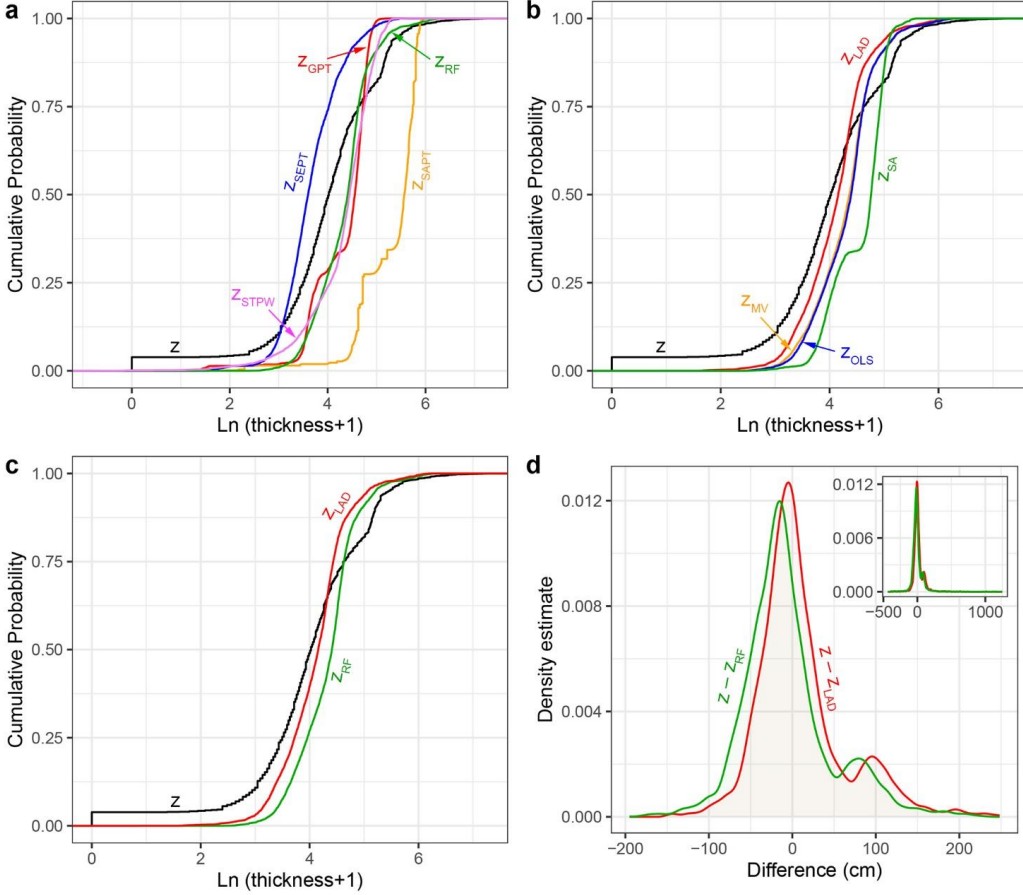

**Figure 11: Cumulative probability of the field-based thickness measurements (z) against the estimations by single models and combination approaches (a and b, respectively). In panel c, only z, the estimations by the best single model and the best combination approach are visualized. However, panel d represents the difference between z and the estimated thickness by the best single model and the best combination technique. In this figure, z represents thickness, but the subscript refers to the method of estimation. The estimations (n = 6137) for both training and test subsets are visualized. GPT: Geomorphological Pyroclastic Thickness; SAPT: Slope Angle Pyroclastic Thickness; SEPT: Slope Exponential Pyroclastic Thickness; STPW: Stepwise regression; RF: Random Forest; SA: Simple Average; MV: Minimum Variance; OLS: Ordinary Least Squares; LAD: Least Absolute Deviation.**

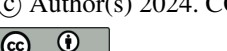

**Figure 12: Spatial visualization of the predicted thickness of fallout pyroclastic deposits by Random Forest as the best single model (the upper panel) and by LAD as the best combination approach (the lower panel) for Vitulano municipality. It is worth mentioning that the Random Forest predictions are classified by the natural breaks method and the thresholds are implemented for generating the map of LAD predictions. Please see the text for more information about A, B and C labels together with the black box.**


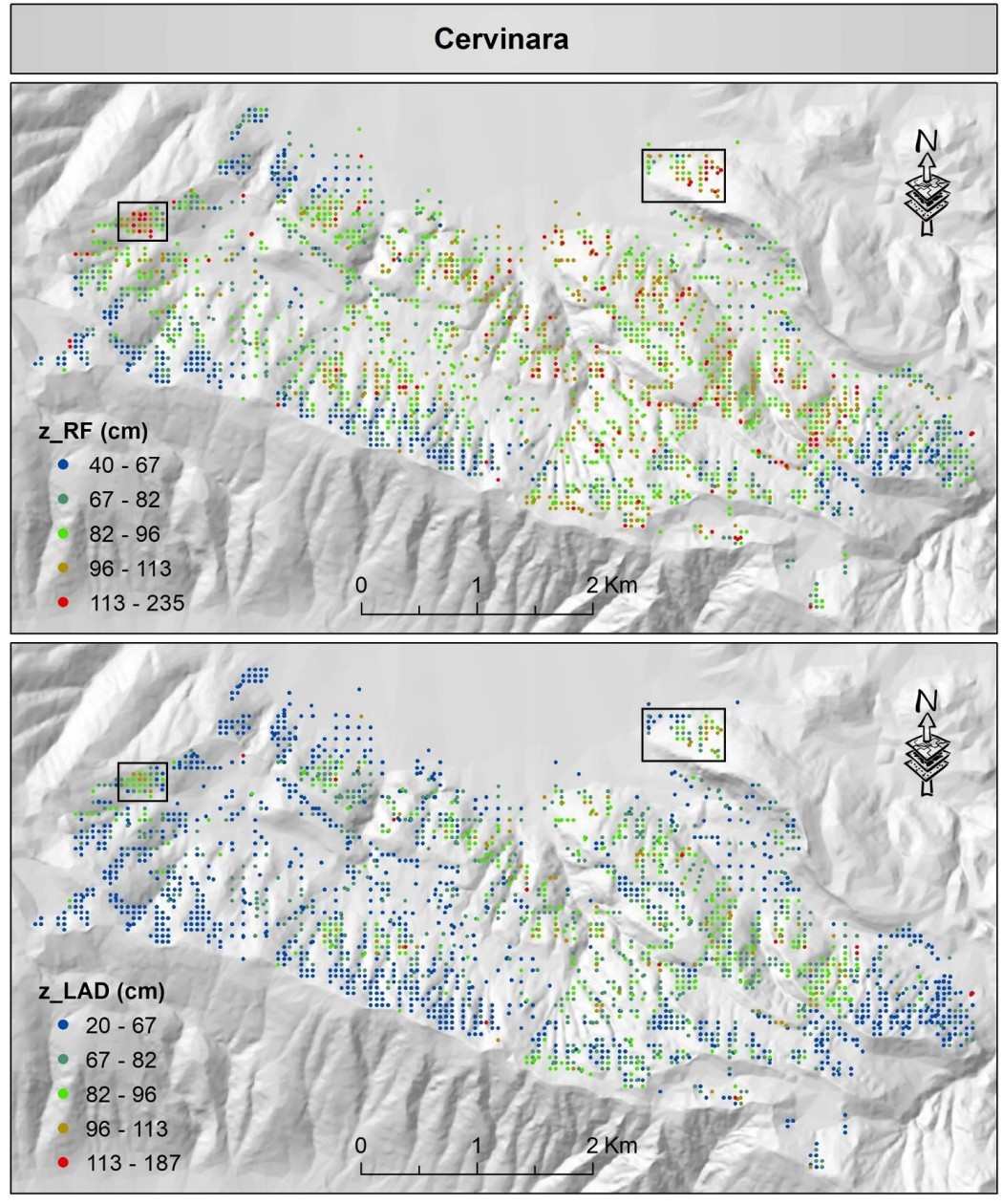

**Figure 13: Spatial visualization of the predicted thickness of fallout pyroclastic deposits by Random Forest as the best single model (the upper panel) and by LAD as the best combination approach (the lower panel) for Cervinara municipality. It is worth mentioning that the Random Rorest predictions are classified by the natural breaks method and the thresholds are implemented for generating the map of LAD predictions. Please see the text for more information about the black boxes.**




**Figure 14: Spatial visualization of the predicted thickness of fallout pyroclastic deposits by Random Forest as the best single model (the upper panel) and by LAD as the best combination approach (the lower panel) for Nocera Inferiore municipality. It is worth mentioning that the RF predictions are classified by the natural breaks method and the thresholds are implemented for generating the map of LAD predictions. Please see the text for more information about the black boxes.**

### 7. Data availability

The field-based thickness measurements of fallout pyroclastic deposits are accessible on Zenodo (Matano et al., 2023; https://doi.org/10.5281/zenodo.8399487).



**8. Conclusion and future research direction**

A given volcano might have several eruptive events. In an explosive volcanic eruption, the height of ash plume and wind characteristics mainly determine the ash-dispersal pattern. However, the expected spatial thickness may be continuously altered by the soil forming and denudation processes. It is, therefore, a daunting task to estimate thickness of fallout pyroclastic deposits that we observe today. The GPT, SAPT and SEPT models were proposed in the previous studies to

address this issue, but this article tries to apply other models for the first time to estimate thickness more accurately around the Somma-Vesuvius, Phlegrean Fields and Roccamonfina volcanoes in Campania region, south Italy.

First, we prepared a database of 6137 field-based thickness measurements with 18 predictor variables. Second, the STPW model and the RF machine learning technique were implemented for thickness modelling and the results were compared with those of GPT, SAPT and SEPT models. The RF estimations (RMSE = 79.11 and MAE = 46.44 for the training

subset, and RMSE = 82.46 and MAE = 48.36 for the test subset) evidently outperformed the other models (RMSE > 89.60 and MAE > 55.25 for the training subset, and RMSE = 92.35 and MAE > 55.20 for the test subset). Third, the SA, LAD, MV and OLS approaches were considered to combine the predictions of the above-mentioned five single models and to obtain more accurate thickness estimations. It was indicated that the LAD approach returns the best results in terms of MAE. Thus, the estimations with the RF and LAD methods (as a single model and a combination approach, respectively)

were less biased in Campania region. The thickness values obtained from the RF and LAD in Vitulano, Cervinara and Nocera Inferiore were applied for spatial analysis and it was demonstrated that the estimated values of the LAD approach are smaller than those of RF, but the spatial patterns do not change significantly. The results showed that following the methodology in this article and generating a map by the estimations of the LAD combination approach provides the most representative estimations in the study area.

In the future, we consider a set of more representative predictor variables (if any) and collect a larger field-based thickness measurement dataset for estimating thickness in the unmeasured locations (i.e. out-of-sample predictions from statistical point of view) more accurately. It would help generate a regional map of thickness of fallout pyroclastic deposits in Campania region which plays a key role in hydrogeological and volcanological studies and in managing geohazards in the areas covered with loose pyroclastic materials. Furthermore, we aim to define the best statistical combinations at local

levels by means of clusterwise techniques.

**Author contribution**

All authors wrote the original draft and revised the manuscript. FM, PE and VA: geology, geomorphology and volcanological history; RM, GS and PE: statistical analyses; FM: Field measurements and supervision of the research activity.

**Competing interests**

The authors declare that they have no conflict of interest.

**Acknowledgements**



Authors acknowledge prof. Paola Petrosino for suggestions about volcanic history, and dr. Massimo Cesarano and dr. Annarita Casaburi for useful comments.

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
