# Peer review of "A field-based thickness measurement dataset of fallout pyroclastic deposits in the peri-volcanic areas of Campania region (Italy): Statistical combination of different predictions for spatial thickness estimation"

_Earth System Science Data, 2024_

## Author Comment (AC1)

**Responses to the Reviewers and Editor Comments - ESSD-2024-44**

**Topic Editor**

**Comment E-1:** I would recommend that the authors better emphasize the description of the database and the information stored in it, with less emphasis on the analyses.

***Response to Comment E-1****: Thank you for the constructive comment. We address this point in the revised manuscript and welcome any specific suggestion for improving description of the database.*

***Changes in manuscript:*** *This point will be addressed by emphasizing the description of the database and explaining uncertainty of the field-based measurements. Please see the changes related to Comment R2-3 for more information about uncertainty of the field-based measurements.*

**Referee #1**

**Comment R1-1:** The paper presents an interesting dataset about the field measurement of fallout pyroclastic deposits thickness in the peri-volcanic areas of Campania region (Italy). Moreover, the authors discuss a relevant problem that can be solved using the dataset and statistical models, that is the spatial thickness estimation using statistical methods. In particular, the combination of different statistical and geological methods is proposed, and alternative combination schemes are compared. Overall, I find the work clear and the paper to be well written. I found the idea discussed in the paper very interesting, and the statistical procedures proposed for thickness estimation are adequate for dealing with the problem at hand.

***Response to Comment R1-1****: We appreciate the positive feedback and constructive comments.*

**Changes in manuscript:** *Not applicable.*

**Comment R1-2:** However, the description of the dataset is not well organized yet in my view. I think more effort should be devoted presenting relevant information about the data adopted for the analysis. Thus, my main suggestion is to improve Section 4 of the manuscript. I do not think all the figures included in the current version of the paper are relevant, and a better selection of the most interesting ones should be considered.

***Response to Comment R1-2:*** *We thank the reviewer for this suggestion. We improve Section 4 by adding more information about the data, keeping the most relevant figures, and explaining measurement uncertainty.*

***Changes in manuscript:*** *Measurement uncertainty (please see the changes related to Comment R2-3) will be explained in Section 3.1. and not in Section 4 to avoid repetition. In addition, Tables 2 and 3 together with Fig. 10 will be placed in the Appendix/Supplementary Materials.*

**Comment R1-3:** Another minor comment is about Table 7. Consider to replace "constant" with "test statistics" or simply "statistics".

***Response to Comment R1-3:*** *We consider this comment in the revision process.*

*Changes in manuscript: This column will be removed because RMSE and MAE were well introduced in the manuscript. The revised Table 7 will be as follows:*

| Accuracy measure | RF vs. SA | RF vs. MV | RF vs. LAD | RF vs. OLS |
|---|---|---|---|---|
| RMSE | -2076.229$^{***}$ (432.758) | -7.006 (10.328) | -125.269 (126.711) | 5.971 (6.188) |
| MAE | -13.493$^{***}$ (0.953) | 0.390$^{***}$ (0.066) | 3.245$^{***}$ (0.397) | 0.091$^{***}$ (0.013) |

***$p < 0.01$

**Comment R1-4:** Some references seem to be incomplete. Be sure all references have volumes, issue and pages. Include the doi for all the papers if possible.

*Response to Comment R1-4: Thank you for reminding us of this point. We address this comment in the revised manuscript.*

*Changes in manuscript: The reference list will be updated as requested. We do not put the updated reference list below for the sake of brevity.*

**Referee #2**

**Comment R2-1:** The paper deals with the application of statistical tools for reconstructing the original thickness of pyroclastic deposits on mountain slopes. Examples are provided from Campania region, downwind of the main Neapolitan volcanoes. In my opinion the manuscript presents some important flaws that need to be fixed before consideration for publication.

*Response to Comment R2-1: We would like to express our gratitude to the reviewer for raising some constructive comments, encouraging us to improve the manuscript especially in relation to the volcanological terminology and bibliography. Briefly, we implement statistical approaches to estimate thickness of the pyroclastic deposits covering peri-volcanic slopes today for understanding the geomorphological processes at the catchment scale. The detailed reconstruction and analysis of volcanological scenarios are not the main aim of our research. The methodology applied in this manuscript outperformed those available in the literature which shows capability of statistical techniques in answering scientific questions and motivates researchers to implement more advanced statistical methods to bridge the existing knowledge gaps. It does not mean that the methodology is applicable to specific case studies for detailed scale investigations. Please note that the revised manuscript will be improved about volcanological terminology and referencing.*

*Changes in manuscript: The manuscript will be revised based on the interesting comments provided by the reviewer.*

**Comment R2-2:** The first one is the lack of adequate volcanological terminology and referencing, which make it the reading sometimes very hard to understand and to be correctly placed in current state of the art of the volcanological literature.

*Response to Comment R2-2: The recent publications will be taken into consideration to update the manuscript regarding volcanological terminology and referencing. Typos and errors in the text relating to terminology will be carefully reviewed as well.*

*Changes in manuscript: The corrections will be mainly made in Section 2.*

**Comment R2-3:** The second critical point is the absence of any discussion about the uncertainty associated to input data. Some of the methods described in the text (SPT, coring, etc) have associated large errors related to the method itself, which can significantly alter the measured thickness of the pyroclastic deposit. In some cases the bias introduced by measurement method can be comparable with reduction to the original thickness by erosion. It is not clear how the interceding of paleosoils is treated when measures are acquired using penetrometric tests.

*Response to Comment R2-3: We would like to express our gratitude to the reviewer for the constructive comment about measurement uncertainty. An in-depth analysis and discussion of the measurement errors and uncertainty associated with the input data, which will be added to the revised manuscript.*

*Changes in manuscript: The discussion below will be added to section 3.1 (minor modifications might be applied, if required):*

*"The following new Table 1 shows the measurement errors and estimated uncertainties of interpretation for each measurement method as used in the study areas. Thickness of ashfall pyroclastic deposits was directly measured using outcrops, hand-dug pits, trenches and boreholes and the error is expected to be approximately 1 cm, as far also for probing tests. The measurement error of the penetration tests (i.e. Dynamic Cone Penetration test (DPT–DL030) and Standard Penetration Test) is considered 10 cm because the number of blows were counted following driving the rod into the ground for 10 cm. In seismic surveys, the measurement error depends on the specific technique and site characteristics, but a measurement error of 100 cm might be a good estimation for the whole study area.*
*The error and interpretative uncertainty of the measurements (i.e. estimation of the fallout pyroclastic deposits thickness) are equal (i.e. 1 cm) in the methods that enable direct thickness measurements. However, interpretative uncertainty increases in probing tests, penetration tests and seismic surveys. It is noteworthy that the results of these types of tests/surveys were calibrated in field based on the more precise tests made nearby (mostly at 1-10 m distance). The weighted average of errors and uncertainty are under 6 and 19 cm, respectively. Therefore, the bias introduced by measurement errors and interpretative uncertainties are not relevant for our study aims".*

*The presence of paleosoils has been considered during field surveys for acquisition of thickness measures by combining data of penetrometric tests and stratigraphy observed in nearby outcrops, hand-dug pits and trenches. The implication of this is considered in the discussion of revised manuscript.*

**new Table 1** - A summary of the field-based thickness measurements of fallout pyroclastic deposits in the peri-volcanic areas of Campania region (Matano et al., 2023). The expected measurement error and interpretation accuracy for each method are also provided. Please note that Forio and Procida (i.e. 41 measurements) are excluded in this article.

| Measurement method | Number of measurements in | | | | | | | | | Total number of measurements | Percentage of measurements | Measurement error (cm) | Interpretation accuracy (cm) |
|---|---|---|---|---|---|---|---|---|---|---|---|---|---|
| | Bagnoli Irpino | Cervinara | Forio | Gioia Sannitica | Nocera Inferiore | Pozzilli | Procida | Villa Santa Lucia | Vitulano | | | | |
| Borehole | 0 | 0 | 0 | 0 | 3 | 0 | 0 | 0 | 0 | 3 | 0.04 | 1 | 1 |
| Dynamic Cone Penetration tests | 0 | 212 | 0 | 0 | 39 | 13 | 6 | 0 | 22 | 292 | 4.38 | 10 | 30 |
| Hand-dug pit | 22 | 234 | 0 | 0 | 41 | 0 | 0 | 20 | 0 | 317 | 4.75 | 1 | 1 |
| Outcrop | 1 | 0 | 0 | 0 | 13 | 0 | 7 | 0 | 131 | 152 | 2.28 | 1 | 1 |
| Probing test | 393 | 2356 | 28 | 31 | 1188 | 12 | 0 | 64 | 1301 | 5373 | 80.54 | 1 | 10 |
| Seismic survey | 0 | 0 | 0 | 0 | 300 | 0 | 0 | 0 | 0 | 300 | 4.50 | 100 | 200 |
| Standard Penetration Test | 0 | 0 | 0 | 0 | 20 | 0 | 0 | 0 | 0 | 20 | 0.30 | 10 | 30 |
| Trench | 23 | 20 | 0 | 19 | 0 | 0 | 0 | 0 | 152 | 214 | 3.21 | 1 | 1 |
| Total number of measurements | 439 | 2822 | 28 | 50 | 1604 | 25 | 13 | 84 | 1606 | 6671 | | | |
| Percentage of measurements | 6.58 | 42.30 | 0.42 | 0.75 | 24.04 | 0.38 | 0.19 | 1.26 | 24.07 | | | | |
| Weighted average of errors | | | | | | | | | | | | **5.9** | **18.6** |

**Comment R2-4:** The uncertainty of results is even more important if we consider that the original thickness is derived from published isopach maps, which are the results of approximation and interpolation themselves.

*Response to Comment R2-4: We would like to express our gratitude to the reviewer for raising the constructive comments, encouraging us to improve the manuscript in relation to the uncertainty associated with these data. The original thickness of pyroclastic deposits was derived from isopach maps, which have been published in peer-reviewed international journals and validated by the scientific community. An analysis of related uncertainty, discussed in the original publications about isopach maps, is reported in the revised manuscript and considered in the discussions.*

*As a preliminary remark, it is noted that in most of the scientific publications describing the isopach maps, the uncertainty regarding the original thickness reported in the maps for specific eruption or for cumulate thickness due to several eruptions is not considered nor discussed both for Somma-Vesuvius and for Campi Flegrei (see De Vita et al., 2006; De Vita & Nappi, 2013; Del Soldato et al., 2016, 2018; Di Vito et al., 2008; Cappelletti et al., 2003; Costa et al., 2009; Isaia et al., 2004; Orsi et al., 2009; Rolandi et al., 2003, 2004, 2007, 2008). Lirer et al. (2001), Orsi et al. (2004) and De Vita et al. (1999) wrote that the isopach maps for fallout deposits have been processed from a minimum thickness of 10 cm as thinner beds usually are not precisely measurable in the field. Costa et al (2012) modeled the IC isopachs best fitting on 113 measures, and the modeled results are in general agreement with the measured thicknesses, the relative mean error is approximately 0.3 log-units.*

*Uncertainty of the statistical results is considered in several sections of the manuscript. In "3.3.4. Accuracy evaluation", we present the statistical indices used to evaluate accuracy of the models: Root Mean Square Error (RMSE, formula 10) and Mean Absolute Error (MAE, formula 11).*

*In Section 5.2, RMSE and MAE for all statistical models are presented, considering both training and test subsets (see Table 5). It shows relevant differences in the accuracy of the models (e.g. MAE=55.25 for STPW vs. MAE=46.44 for RF).*

*To account for the uncertainty in the results while evaluating the differences of the models, we consider statistical tests called "predictive accuracy tests". In the revised manuscript, we will include the following new Table 2 showing the predictive accuracy tests' results between the single statistical models included in Table 5, for the test subset.*

**new Table 2.** Equal predictive accuracy tests applied to the test subset (n = 1843), comparing the best single model (RF) with other single approaches. Under the null hypothesis, the two models under comparison provide equal predictive accuracy. The heteroskedasticity robust standard errors are in parentheses.

| Prediction error loss | RF vs. GPT | RF vs. SEPT | RF vs. SAPT | RF vs. STPW |
|---|---|---|---|---|
| Squared errors | -2650.0[***] (705.2) | -28523.4[***] (753.6) | -4745.0[***] (821.5) | -1729.0[***] (533.6) |
| Absolute errors | -8.564[***] (1.113) | -111.94[***] (2.2730) | -9.689[***] (1.4320) | -6.8339[***] (0.9109) |

*The results show that the differences between the RF and all the other models are statistically significant. Said differently, we find that the observed differences in the models' performance are not explained by randomness of the data only. Furthermore, we statistically compare the best single model (i.e. RF) in terms of accuracy with the different combination models in Table 7. We see that, accounting for the results' uncertainty, the combination models give more accurate statistical predictions than RF. In the revised manuscript we also will better explain why predictive accuracy tests account for the uncertainty of the results.*

***Changes in manuscript:*** *Accuracy of the combination models will be listed in the new table 2 and an explanation of the predictive accuracy tests will be included.*

**Comment R2-5:** Line 101: The somma Vesuvius summit caldera is the result of the 4 main Pinian eruption of the volcano. The AD 79 is only the last one (Cioni et al., 1999; Santacroce et al., 2008). I wonder which eruption is 18 AD. The references cited are not appropriate, because of the mess of publications regarding the eruptive activity of SV.

***Response to Comment R2-5:*** *It is highly appreciated that you reminded this point. The sentence will be appropriately revised to make sense from the volcanological point of view.*

***Changes in manuscript:*** *The morpho-structure of the Somma-Vesuvius volcano is better clarified in the lines 154-158. Also, more appropriate references will be added.*

**Comment R2-6:** Line 103: The term "large explosively index" has no meaning. Please use more appropriate terminology

***Response to Comment R2-6:*** *Thank you for the suggestion.*

***Changes in manuscript:*** *The term will be changed as "large and very large Volcanic Explosivity Index (VEI; Newhall and Self, 1982)".*

Newhall, C. G., & Self, S. (1982). The volcanic explosivity index (VEI) an estimate of explosive magnitude for historical volcanism. *Journal of Geophysical Research: Oceans*, 87(C2), 1231-1238. https://doi.org/10.1029/JC087iC02p01231

**Comment R2-7:** Line 104: the activity of Phlegran Fields did not initiate with the CI (see Orsi et al., 1996; Di Vito et al., 2008)

***Response to Comment R2-7:*** *Thank you for notifying this. The sentence will be changed.*

***Changes in manuscript:*** *The sentence "The beginning of Phlegrean volcanism is not well constrained yet, but it is well known that activity of the volcanic field started from the super-eruption of the Campanian Ignimbrite" will be changed as "Volcanic activity in the Phlegrean Fields began prior to 80 ka BP (Pappalardo et al., 1999; Scarpati et al., 2013) and the caldera collapses occurred during the eruptions of Campanian Ignimbrite (ca. 39ka BP; Deino et al., 1994; De Vivo et al., 2001), Masseria del Monte Tuff (29 ka BP; Albert et al., 2019) and Neapolitan Yellow Tuff (NYT: 15 ka BP; Orsi et al., 1996; Perrotta et al., 2006; Acocella, 2008; Vitale and Isaia, 2014). The post-NYT activity in the caldera is well described by Di Vito et al. (1999), Isaia et al. (2009) and Smith et al. (2011)."*

**Comment R2-8:** I wonder why you introduced Ischia and Roccamonfina volcanoes if you do not use their deposits in the manuscript.

***Response to Comment R2-8:*** *Good point. Some field-based measurements around Roccamonfina (e.g., Pozzilli area) were included in the manuscript databaset, but those in Procida and Forio were excluded by statistical analyses because only little data was available. Thus, they are introduced in the manuscript to provide a general overview of the dataset.*

***Changes in manuscript:*** *Not applicable.*

**Comment R2-9:** The volcanic history of Phlegrean Fields may be shortened and better described.

*Response to Comment R2-9: Agree*

*Changes in manuscript: Section 2.1 related to the eruptive history of Phlegrean Field will be shortened and better described in the revised manuscript.*

**Comment R2-10:** The same for the SV eruptive history. Please, cite also the large amount of newer literature available for most of the cited eruptions.

*Response to Comment R2-10: Thank you. This constructive comment will be considered during revision. The subsection related to the eruptive history of Somma-Vesuvius will be shortened and better described regarding the recently published articles.*

*Changes in manuscript: Section 2.2 will be updated as requested by the reviewer.*

**Comment R2-11:** Line 44: please provide more references. The cited authors are not the first that noted the thickness decrease with distance, which is a common sense in volcanology.

*Response to Comment R2-11: Other references will be provided to support the statement.*

*Changes in manuscript: The following references will be added to address this comment:*

Lowe, D. J. (2011). Tephrochronology and its application: a review. Quaternary Geochronology, 6(2), 107-153. https://doi.org/10.1016/j.quageo.2010.08.003

Bourne, A. J., Lowe, J. J., Trincardi, F., Asioli, A., Blockley, S. P. E., Wulf, S., ... & Vigliotti, L. (2010). Distal tephra record for the last ca 105,000 years from core PRAD 1-2 in the central Adriatic Sea: implications for marine tephrostratigraphy. Quaternary Science Reviews, 29(23-24), 3079-3094. https://doi.org/10.1016/j.quascirev.2010.07.021

Brown, R. J., Bonadonna, C., & Durant, A. J. (2012). A review of volcanic ash aggregation. Physics and Chemistry of the Earth, Parts a/b/c, 45, 65-78. https://doi.org/10.1016/j.pce.2011.11.001

Caron, B., Siani, G., Sulpizio, R., Zanchetta, G., Paterne, M., Santacroce, R., ... & Zanella, E. (2012). Late Pleistocene to Holocene tephrostratigraphic record from the northern Ionian Sea. Marine Geology, 311, 41-51. https://doi.org/10.1016/j.margeo.2012.04.001

Costa, A., Folch, A., Macedonio, G., Giaccio, B., Isaia, R., & Smith, V. C. (2012). Quantifying volcanic ash dispersal and impact of the Campanian Ignimbrite super-eruption. Geophysical Research Letters, 39(10). https://doi.org/10.1029/2012GL051605

Eychenne, J., & Engwell, S. L. (2023). The grainsize of volcanic fall deposits: Spatial trends and physical controls. GSA Bulletin, 135(7-8), 1844-1858. https://doi.org/10.1130/B36275.1

**Comment R2-12:** Line 102: Phlegrean Fields are not a volcanic field but a caldera.

*Response to Comment R2-12: The correction will be applied in the revised manuscript.*

*Changes in manuscript: The sentence "The Phlegrean Fields refer to a volcanic field located immediately westward Naples" will be fully rephrased as "The Phlegrean Fields consist of several volcanoes located in a large caldera located westward Naples".*

**Comment R2-13:** Line 129: I wonder what means "under 80 km from the eruptive vent".

*Response to Comment R2-13: Thank you for informing us that this part is not clear.*

*Changes in manuscript: The sentence "The outcrops associated with this eruption are found in the Campanian Plain and the Apennine chain (under 80 km from the eruptive vent), mainly composed of a stratified pumice deposit overlay a grey welded tuff unit" will be revised as "Both fallout and pyroclastic density current (PDC) deposits were emplaced by the CI eruption (Barberi et al., 1978; Rosi et al., 1988; Fisher et al., 1993; Orsi et al., 1996; Rosi et al., 1996; De Vivo et al., 2001; Cappelletti et al., 2003; Fedele et al., 2008; Engwell et al., 2014; Scarpati et al., 2015; Smith et al., 2016). The former was occurred during the first eruptive phase and dispersed by winds towards the east (Rosi et al., 1999; Perrotta and Scarpati, 2003; Marti et al., 2016; Scarpati and Perrotta, 2016), arriving up to the Eastern Mediterranean Sea and Eastern Eurasia (Giaccio et al., 2014; Smith et al., 2016). The latter was subsequently emplaced over an area of 7000 km$^2$ and surmounted ridges more than 1000 m high (Barberi et al., 1978; Fisher et al., 1993). The CI distal outcrops are mostly represented by a massive, grey ignimbrite (Barberi et al., 1978; Fisher et al., 1993; Scarpati et al., 2015), distributed beyond ~80 km from the vent (Smith et al., 2016).*

**Comment R2-14:** Line 130: please rephrase. It has not meaning as it is.

*Response to Comment R2-14: Thank you for notifying this point.*

*Changes in manuscript: The sentence will be rephrased.*

**Comment R2-15:** Table 3: Taurano is not a Phlegran Fields eruption (Di Vito et al., 2008)

*Response to Comment R2-15: Apologies for including Taurano in the wrong table.*

*Changes in manuscript: Taurano eruption consist of several fall deposits occurred between 33 and 36 cal ka BP (Di Vito et al., 2008) from Somma-Vesuvius. For this reason, this eruption will be moved from Table 3 to Table 2 (note: these tables will be placed in the Appendix/Supplementary materials as requested by the reviewer 1).*

---

## Author Response (AR1)

**Responses to the Reviewers and Editor Comments - ESSD-2024-44**

**Topic Editor**

**Comment E-1:** I would recommend that the authors better emphasize the description of the database and the information stored in it, with less emphasis on the analyses.

*Response to Comment E-1: Thank you for the constructive comment. We addressed this point in the revised manuscript.*

*Changes in manuscript: We tried to improve the manuscript by explaining uncertainty of the field-based thickness measurements and discussing the database more. Please see sections 3.1, 3.2 and 4 along with the changes related to Comment R2-3.*

**Referee #1**

**Comment R1-1:** The paper presents an interesting dataset about the field measurement of fallout pyroclastic deposits thickness in the peri-volcanic areas of Campania region (Italy). Moreover, the authors discuss a relevant problem that can be solved using the dataset and statistical models, that is the spatial thickness estimation using statistical methods. In particular, the combination of different statistical and geological methods is proposed, and alternative combination schemes are compared. Overall, I find the work clear and the paper to be well written. I found the idea discussed in the paper very interesting, and the statistical procedures proposed for thickness estimation are adequate for dealing with the problem at hand.

*Response to Comment R1-1: We appreciate the positive feedback and constructive comments.*

**Changes in manuscript:** *Not applicable.*

**Comment R1-2:** However, the description of the dataset is not well organized yet in my view. I think more effort should be devoted presenting relevant information about the data adopted for the analysis. Thus, my main suggestion is to improve Section 4 of the manuscript. I do not think all the figures included in the current version of the paper are relevant, and a better selection of the most interesting ones should be considered.

*Response to Comment R1-2: We thank the reviewer for this suggestion. Relevant information was added to the manuscript.*

*Changes in manuscript: The dataset was further discussed in Section 4. Measurement uncertainty (please also see the changes related to Comment R2-3) has also been explained in Section 3.1 and not in Section 4 to avoid repetition. The uncertainty and accuracy of the predictor variables were further explained in section 3.2. In addition, Tables 2 and 3 together with Fig. 10 were placed in the Supplementary Material.*

**Comment R1-3:** Another minor comment is about Table 7. Consider to replace "constant" with "test statistics" or simply "statistics".

*Response to Comment R1-3: We considered this comment in the revision process.*

*Changes in manuscript: This column was removed because RMSE and MAE were well introduced in the manuscript. The table was also updated to address other comments as follows:*

*Table 6. Equal predictive accuracy tests, applied to the test subset (n = 1843), for comparing the best single model (RF) with the other ones. Under the null hypothesis, the two models provide equal predictive accuracy. SE refers to the heteroskedasticity robust standard error. The abbreviations are as Table 4. \*\*\* indicates p-value<0.01.*

| Best single model | Other models | RMSE | SE | MAE | SE |
|---|---|---|---|---|---|
| RF vs. | GPT | -2650.0*** | 705.2 | -8.564*** | 1.113 |
| | SAPT | -4745.0*** | 821.5 | -9.689*** | 1.4320 |
| | SEPT | -28523.4*** | 753.6 | -111.94*** | 2.2730 |
| | STPW | -1729.0*** | 533.6 | -6.8339*** | 0.9109 |
| | SA | -2076.229*** | 432.758 | -13.493*** | 0.953 |
| | MV | -7.006 | 10.328 | 0.390*** | 0.066 |
| | OLS | 5.971 | 6.188 | 0.091*** | 0.013 |
| | LAD | -125.269 | 126.711 | 3.245*** | 0.397 |

**Comment R1-4:** Some references seem to be incomplete. Be sure all references have volumes, issue and pages. Include the doi for all the papers if possible.

*Response to Comment R1-4: Thank you for reminding us of this point. We have addressed this comment in the revised manuscript.*

*Changes in manuscript: The reference list has been updated as requested. We do not put the updated reference list below for the sake of brevity.*

**Referee #2**

**Comment R2-1:** The paper deals with the application of statistical tools for reconstructing the original thickness of pyroclastic deposits on mountain slopes. Examples are provided from Campania region, downwind of the main Neapolitan volcanoes. In my opinion the manuscript presents some important flaws that need to be fixed before consideration for publication.

*Response to Comment R2-1: We would like to express our gratitude to the reviewer for raising some constructive comments, encouraging us to improve the manuscript especially in relation to the volcanological terminology and bibliography. Briefly, we implemented statistical approaches in the manuscript to estimate thickness of the pyroclastic deposits covering peri-volcanic slopes today for understanding the geomorphological processes at the catchment scale. The detailed reconstruction and analysis of volcanological scenarios are not the main aim of our research. The methodology applied in this manuscript outperformed those available in the literature which shows capability of the statistical techniques in answering scientific questions and motivates researchers to implement more advanced statistical methods to bridge the existing knowledge gaps. It does not mean that the methodology is applicable to specific case studies for detailed scale investigations.*

*Changes in manuscript: The revised manuscript has been improved on volcanological terminology, citations and measurement uncertainty. Please see the responses to the comments.*

**Comment R2-2:** The first one is the lack of adequate volcanological terminology and referencing, which make it the reading sometimes very hard to understand and to be correctly placed in current state of the art of the volcanological literature.

**Response to Comment R2-2:** *The manuscript has been carefully revised.*

**Changes in manuscript:** *The following citations were added to the text and the reference list was updated:*

*Barberi et al., 1978; Bourne et al., 2010; Brown et al., 2012; Caron et al., 2012; Cioni et al., 2015; Doronzo et al., 2022, 2023; Engwell et al., 2014; Eychenne and Engwell, 2023; Isaia et al., 2009; Lowe, 2011; Mele et al., 2011; Newhall and Self, 1982; Pappalardo et al., 1999; Perrotta et al., 2006; Rosi et al., 1999; Sbrana et al., 2020; Scarpati and Perrotta, 2016; Scarpati et al., 2014, 2015; Sigurdsoon et al., 1983; Smith et al., 2016; Vitale and Isaia, 2014*

*In addition, the discussion on volcanic activity of Phlegrean Fields and Somma-Vesuvius was revised. Please see changes in manuscript related to Comment R2-9 and Comment R2-10.*

**Comment R2-3:** The second critical point is the absence of any discussion about the uncertainty associated to input data. Some of the methods described in the text (SPT, coring, etc) have associated large errors related to the method itself, which can significantly alter the measured thickness of the pyroclastic deposit. In some cases the bias introduced by measurement method can be comparable with reduction to the original thickness by erosion. It is not clear how the interceding of paleosoils is treated when measures are acquired using penetrometric tests.

**Response to Comment R2-3:** *We would like to express our gratitude to the reviewer for the constructive comment. An in-depth analysis and discussion of the measurement errors and uncertainty associated with the input data have been added to section 3.1. Paleosols have been recognized during field surveys for thickness acquisition by combining data of outcrops, hand-dug pits and trenches with penetration test results. However, the stratigraphy of the pyroclastic deposits and the possible presence of non-lithified paleosols are not considered in detail in our analyses as they are beyond the scope of this article.*

**Changes in manuscript:** *The following discussion on measurement uncertainty has been added to section 3.1:*

*For each method, the measurement error and estimated interpretation uncertainty (i.e. estimation of the fallout pyroclastic deposit thickness) are shown in Table 1. About 1 cm error is expected for direct thickness measurements through outcrops, hand-dug pits, trenches and boreholes. The error associated with the probing test results is around 1 cm as well. The measurement error of the penetration tests (i.e. Dynamic Cone Penetration test (DPT–DL030) and Standard Penetration Test) is considered 10 cm because the number of blows was counted following driving the rod into the ground for 10 cm. In seismic surveys, the error depends on the specific technique and site characteristics, but a measurement error of 100 cm might be a good estimation for the whole study area.*

*The interpretative uncertainty of the measurements is equal to the measurement error (i.e. 1 cm) for direct thickness measurements, but increases in probing tests, penetration tests and seismic surveys. It is noteworthy that the results of these tests/surveys were calibrated in the field based on the more precise tests nearby (mostly at 1-10 m distance). The weighted average of errors and uncertainty are under 6 and 19 cm, respectively (Table 1). Therefore, the bias introduced by measurement errors and interpretative uncertainties is irrelevant to the objectives of this article.*

*Table 1: The expected measurement error and interpretation uncertainty for the methods implemented.*

| Measurement method | Numer of measurements | Percentage of measurements | Measurement error (cm) | Interpretation uncertainty (cm) |
|---|---|---|---|---|

| | | | | |
|---|---|---|---|---|
| Borehole | 3 | 0.04 | 1 | 1 |
| Dynamic Cone Penetration test | 292 | 4.38 | 10 | 30 |
| Hand-dug pit | 317 | 4.75 | 1 | 1 |
| Outcrop | 152 | 2.28 | 1 | 1 |
| Probing test | 5373 | 80.54 | 1 | 10 |
| Seismic survey | 300 | 4.50 | 100 | 200 |
| Standard Penetration Test | 20 | 0.30 | 10 | 30 |
| Trench | 214 | 3.21 | 1 | 1 |
| Total | 6671 | 100 | | |
| Weighted average of errors | | | 5.9 | 18.6 |

*In addition, the following discussion on stratigraphy and presence of paleosol has been added to the same section:*

*The measurements explain the distance between the topographic surface and the upper limit of the consolidated basement, referring to "total" measurement when the instrument could measure the whole distance and "partial" measurement when limitations of the implemented instrument led to measuring the distance partially. Further details about the stratigraphy of the pyroclastic deposits and the possible presence of non-lithified paleosols are not considered as they are beyond the scope of this article.*

*Uncertainty of the input variables has been discussed below Table 2 and in section 3.2.2:*

*Other variables, such as the distance to the hydrographic network and distance to the source (i.e. eruptive vent) are also considered as predictor variables in this study. In the latter, several inferred eruptive vents reported by Di Vito et al. (2008) along with Vesuvius crater and Roccamonfina caldera were considered to take into account different ash-producing eruptions and reduce the associated uncertainty as far as possible. Further information is provided in Table 2.*

*Vertical accuracy of the DEM is evaluated by control points and the overall root mean square error is <3.5 m (Tarquini et al., (2007).*

*Landsat 8 OLI data are calibrated to better than 5% uncertainty in terms of Top of Atmosphere reflectance and have an absolute geodetic accuracy better than 65 m circular error at 90% confidence (Ihlen, 2019).*

**Comment R2-4:** The uncertainty of results is even more important if we consider that the original thickness is derived from published isopach maps, which are the results of approximation and interpolation themselves.

***Response to Comment R2-4:*** *We would like to express our gratitude to the reviewer for raising the constructive comments, encouraging us to clarify uncertainty of the data. The original thickness of pyroclastic deposits was derived from isopach maps, which have been published in the literature and validated by the scientific community. An analysis of uncertainty, discussed in the original publications about isopach maps is added to the revised manuscript and discussed in section 3.2.1. Uncertainty of the statistical results is considered in several sections of the manuscript. In "3.3.5. Accuracy evaluation", we present the statistical indices used to evaluate accuracy of the models: Root Mean Square Error (RMSE, formula 10) and Mean Absolute Error (MAE, formula 11). In Section 5.2, RMSE and MAE for all statistical models are presented, considering both training and test subsets (see Table 4). To account for the uncertainty in the results while evaluating the differences*

*between models, we consider "equal predictive accuracy tests". In the revised manuscript, discussion on the results of this test was improved.*

***Changes in manuscript:** The discussion below has been added to section 3.2.1.*

*Isopach maps are commonly used in volcanological studies to estimate the volume of a single eruptive event and assess volcanic hazard. They are constructed by interpolating spatial thickness data points, considered reliable as directly measured by investigating the stratigraphy of volcanic deposits. To the best of our knowledge, uncertainty of the published isopach maps for Somma-Vesuvius and Phlegrean Fields (De Vita et al., 1999; Di Vito et al., 2008; Cappelletti et al., 2003; Costa et al., 2009; Isaia et al., 2004; Orsi et al., 2004; Rolandi et al., 2003, 2004, 2007, 2008) has not been discussed in the literature. Only Costa et al. (2012) modeled the Campanian Ignimbrite isopach map based on 113 measurements and reported that the results are in agreement with the measured thickness values (relative mean error= ~0.3 log-units). The uncertainty is neither quantified for the cumulative isopach maps of multiple eruptions generated for studying erosional processes and landslide susceptibility (De Vita et al., 2006a; De Vita &and Nappi, 2013; Del Soldato et al., 2016, 2018).*

*The following table was also updated and interpreted to consider uncertainty in the results and investigate whether the differences in predictive performance are statistically significant in the test subset.*

*Table 6. Equal predictive accuracy tests, applied to the test subset (n = 1843), for comparing the best single model (RF) with the other ones. Under the null hypothesis, the two models provide equal predictive accuracy. SE refers to the heteroskedasticity robust standard error. The abbreviations are as Table 4. \*\*\* indicates p-value<0.01.*

| Best single model | Other models | RMSE | SE | MAE | SE |
|---|---|---|---|---|---|
| RF vs. | GPT | -2650.0*** | 705.2 | -8.564*** | 1.113 |
| | SAPT | -4745.0*** | 821.5 | -9.689*** | 1.4320 |
| | SEPT | -28523.4*** | 753.6 | -111.94*** | 2.2730 |
| | STPW | -1729.0*** | 533.6 | -6.8339*** | 0.9109 |
| | SA | -2076.229*** | 432.758 | -13.493*** | 0.953 |
| | MV | -7.006 | 10.328 | 0.390*** | 0.066 |
| | OLS | 5.971 | 6.188 | 0.091*** | 0.013 |
| | LAD | -125.269 | 126.711 | 3.245*** | 0.397 |

*Finally, pairwise EPA tests are applied to account for the uncertainty in the results and investigate whether the differences in predictive performance are statistically significant in the test subset (Table 6). The upper part of Table 6 compares RF as the best single model with the other single models and reveals that the differences are statistically significant. In other words, the observed differences in predictive performance of the models are not explained by randomness of the data, and RF model provides the most accurate thickness estimation. The negative RMSE and MAE values indicate that the RF model has a lower average prediction error than the other single models. On the other hand, the lower part of Table 6 shows the pairwise comparison of the RF model with all the combination approaches. A positive RMSE or MAE value suggests that the predictions of the combination approach are more accurate. In terms of squared error loss, the OLS combination approach provides the most accurate predictions than the RF model, but it is noteworthy that RMSE and OLS are both sensitive to data outliers as explained in Section 3.3. Regarding the MAE, statistically significant improvement is observed when combination approaches are applied (except for the SA method; p < 0.01). The greatest statistically significant constant value of the LAD method demonstrates that this*

*combination technique is suitable for predicting thickness of fallout pyroclastic deposits in unmeasured locations. The results are in accordance with those in Table 4.*

**Comment R2-5:** Line 101: The somma Vesuvius summit caldera is the result of the 4 main Plinian eruption of the volcano. The AD 79 is only the last one (Cioni et al., 1999; Santacroce et al., 2008). I wonder which eruption is 18 AD. The references cited are not appropriate, because of the mess of publications regarding the eruptive activity of SV.

***Response to Comment R2-5:*** *It is highly appreciated that you reminded this point. The sentence will be appropriately revised to make sense from the volcanological point of view.*

***Changes in manuscript:*** *The relevant sentences in section 2 were revised as follows:*

*The Somma-Vesuvius volcanic complex lies over a large sedimentary plain, prevalently filled by pyroclastic deposits. In this volcanic complex, the older Mt. Somma stratovolcano was cut by an eccentric polyphasic caldera and by the Vesuvius stratocone (Sbrana et al., 2020). Four main Plinian (Cioni et al., 2003; Santacroce et al., 2008; Sulpizio et al., 2010a,b; Mele et al., 2011; Sevink et al., 2011; Doronzo et al., 2022) and several interplinian eruptions (Andronico and Cioni, 2002; Cioni et al., 2015; Sulpizio et al., 2005, 2007; Bertagnini et al., 2006) have been linked to the volcanic activities of Somma-Vesuvius.*

**Comment R2-6:** Line 103: The term "large explosively index" has no meaning. Please use more appropriate terminology

***Response to Comment R2-6:*** *Thank you for the suggestion, but the term we used was "large explosivity (not explosively) index".*

***Changes in manuscript:*** *The sentence was revised as:*
*The Phlegrean Fields consist of several volcanoes in a large caldera westward Naples, characterized by many eruptions with a large and very large Volcanic Explosivity Index (VEI; Newhall and Self, 1982) (Fig. 2).*

**Comment R2-7:** Line 104: the activity of Phlegran Fields did not initiate with the CI (see Orsi et al., 1996; Di Vito et al., 2008)

***Response to Comment R2-7:*** *Thank you for notifying this. The sentence has been rewritten.*

***Changes in manuscript:*** *The bullet point was revised as follows:*
*The Phlegrean Fields consist of several volcanoes in a large caldera westward Naples, characterized by many eruptions with a large and very large Volcanic Explosivity Index (VEI; Newhall and Self, 1982) (Fig. 2). Volcanic activity in Phlegrean Fields began prior to 80 ka BP (Pappalardo et al., 1999; Scarpati et al., 2014) and the caldera collapses occurred during the eruptions of Campanian Ignimbrite (ca. 39 ka BP; Deino et al., 2004; De Vivo et al., 2001), Masseria del Monte Tuff (29 ka BP; Albert et al., 2019) and Neapolitan Yellow Tuff (15 ka BP; Orsi et al., 1996; Perrotta et al., 2006; Vitale and Isaia, 2014). The post-15ka activity was well described by Di Vito et al. (1999), Isaia et al. (2009) and Smith et al. (2011).*

**Comment R2-8:** I wonder why you introduced Ischia and Roccamonfina volcanoes if you do not use their deposits in the manuscript.

**Response to Comment R2-8:** *Good point. Some field-based measurements around Roccamonfina (e.g., Pozzilli area) were included in the manuscript database, but those in Procida and Forio were excluded for statistical analyses because limited data was available. Thus, they are introduced in the manuscript to provide a general overview of the dataset.*

**Changes in manuscript:** *Not applicable.*

**Comment R2-9:** The volcanic history of Phlegrean Fields may be shortened and better described.

**Response to Comment R2-9:** *Agree*

**Changes in manuscript:** *The eruptive history of Phlegrean Field have been shortened and better described in the revised manuscript:*

*The most important volcanic activities in Phlegrean Fields refer to the Campanian Ignimbrite (CI: 39 ka BP; De Vivo et al., 2001) and the Neapolitan Yellow Tuff eruptions (15 ka BP, Orsi et al., 1992, 1995; Wohletz et al., 1995; Deino et al., 2004). The former is the most powerful volcanic event ever occurred in the Mediterranean area (Barberi et al., 1978; Fisher et al., 1993; Orsi et al., 1996; Rosi et al., 1988, 1996; Civetta et al., 1997; De Vivo et al., 2001; Cappelletti et al., 2003; Engwell et al., 2014; Scarpati et al., 2015; Smith et al., 2016) that emplaced thick sequences of fallout deposits and pyroclastic density currents of mostly trachytic composition (Giaccio et al, 2008; Costa et al, 2022). The dispersed ash during the first eruptive phase was transported by wind toward east (Rosi et al., 1999; Perrotta and Scarpati, 2003; Scarpati and Perrotta, 2016). The pyroclastic density currents was subsequently emplaced over an area of 7,000 $km^2$ and surmounted ridges with more than 1,000 m high (Barberi et al., 1978; Fisher et al., 1993). The CI distal outcrops are mostly represented by a massive, grey ignimbrite (Barberi et al., 1978; Fisher et al., 1993; Scarpati et al., 2014), distributed beyond ~80 km from the vent (Smith et al., 2016).*

*The post-15ka activity of Phlegrean Fields was concentrated in three epochs separated by two quiescent periods (Fig. 2; Di Vito et al., 1999; Smith et al., 2011; Di Renzo et al., 2011 and references therein), and terminated with the Monte Nuovo eruption in 1538 CE (Guidoboni and Ciuccarelli, 2011; Di Vito et al., 2016 and references therein). The first epoch (15 to ~9.5 ka BP) is characterized by several explosive events, of which Pomici Principali eruption was the most energetic one (Lirer et al., 1987; Di Vito et al., 1999). This epoch was followed by a quiescent period when a thick paleosol layer, pedomarker A, was developed. The second epoch (8.6-8.2 ka BP; Di Vito et al., 1999) is distinguished by only a few episodes of low-magnitude eruptions mainly in NE Campanian Plain. After pedomarker B formation in a prolonged volcanic quiescence, the last epoch of intense volcanic activity began between 4.4 and 3.8 ka BP (Di Vito et al., 1999). The third epoch is characterized by several explosive events, of which the Agnano-Monte Spina eruption (4.4 ka BP; De Vita et al., 1999; Dellino et al., 2001) was the most powerful. This epoch was followed by a prolongate quiescent period and Monte Nuovo eruption (1538 CE; Di Vito et al., 1987; Piochi et al., 2005), respectively. Since 1960, fumarolic and hydrothermal activities with episodes of bradyseism mainly occur in Phlegrean Fields (Cannatelli et al., 2020).*

**Comment R2-10:** The same for the SV eruptive history. Please, cite also the large amount of newer literature available for most of the cited eruptions.

**Response to Comment R2-10:** *Thank you. The subsection related to the eruptive history of Somma-Vesuvius was shortened and better described regarding the recently published articles.*

**Changes in manuscript:** *Section 2.2 has been updated as requested by the reviewer:*

*The Somma-Vesuvius volcanic activity is characterized by four major Plinian eruptions (i.e. Pomici di Base or "Sarno" at ca. 22 ka BP, Mercato or "Ottaviano" at ca. 9.0 ka BP, Avellino at 3.9 ka BP and Pompeii at 79 CE) and several low-intensity interplinian eruptions (Fig. 2). Pomici di Base (Andronico et al., 1995; Santacroce et al., 2008) was the oldest caldera-forming event, which was followed by notably variable interplinian activities, alternating low-magnitude eccentric flank eruptions, quiescent phases and subplinian events (such as the Greenish Pumice eruption at ~19 ka BP; Santacroce and Sbrana, 2003; Santacroce et al., 2008). The products of Mercato eruption (Rolandi et al., 1993a; Mele et al., 2011) occurred about thirteen thousand years later were separated from those of Avellino eruption (Rolandi et al., 1993b; Sulpizio et al., 2010a,b; Sevink et al., 2011) by a thick paleosol layer (Di Vito et al., 1999). The low-intensity eruptions of AP1-AP6 (3.5-2.3 ka BP; Andronico et al., 2002; Santacroce et al., 2008; Passariello et al., 2010; Di Vito et al., 2019) preceded eruption of Pompeii which was well described by many authors (from Sigurdsoon et al. 1983 to Doronzo et al. 2023 and references therein).*

*The Vesuvius cone was formed by the most recent period of volcanic activity, characterized by a complex alternation of periods of activity with various explosive characters and quiescent phases (Andronico et al., 1995), suddenly interrupted by the Pollena eruption (472 CE; Rolandi et al., 2004). A Middle Age period of variable activity was then started, alternating lava effusions, moderately explosive eruptions, and mild periods (Rolandi et al., 1998), before a subplinian eruption in 1631 CE (Bertagnini et al., 2006). After this event, the volcano entered a semipersistent mild activity with minor lava effusions and short quiescent periods. Each period of repose was preceded by relatively powerful explosive and effusive polyphase eruptions (Arrighi et al., 2001) like the last two ones in 1908 and 1944.*

**Comment R2-11:** Line 44: please provide more references. The cited authors are not the first that noted the thickness decrease with distance, which is a common sense in volcanology.

*Response to Comment R2-11: Other references have been provided to support the statement.*

*Changes in manuscript: The revised sentence is as follows:*

*Spatial thickness of the ash layer typically decreases with distance from the eruptive vent (e.g., see Perrotta and Scarpati, 2003; Bourne et al., 2010; Lowe, 2011; Brown et al., 2012; Caron et al., 2012; Costa et al., 2012; Albert et al., 2019; Eychenne and Engwell, 2023) and noticeably influences geomorphological and hydrological processes such as landscape evolution, hillslope hydrology, erosion, and slope stability because the geotechnical and hydraulic properties of the unconsolidated ash layer usually differ from the underlying bedrock and soil.*

**Comment R2-12:** Line 102: Phlegrean Fields are not a volcanic field but a caldera.

*Response to Comment R2-12: The correction has been applied in the revised manuscript.*

*Changes in manuscript: The sentence is revised as follows:*
*The Phlegrean Fields consist of several volcanoes in a large caldera westward Naples, characterized by many eruptions with a large and very large Volcanic Explosivity Index (VEI; Newhall and Self, 1982) (Fig. 2).*

**Comment R2-13:** Line 129: I wonder what means "under 80 km from the eruptive vent".

*Response to Comment R2-13: Thank you for informing us that this phrase is not clear.*

*Changes in manuscript: The sentence was replaced with the following one:*
*The CI distal outcrops are mostly represented by a massive, grey ignimbrite (Barberi et al., 1978; Fisher et al., 1993; Scarpati et al., 2014), distributed beyond ~80 km from the vent (Smith et al., 2016).*

**Comment R2-14:** Line 130: please rephrase. It has not meaning as it is.

***Response to Comment R2-14:*** *Done.*

***Changes in manuscript:*** *The sentence has been rewritten. Please see the response to comment R2-13.*

**Comment R2-15:** Table 3: Taurano is not a Phlegran Fields eruption (Di Vito et al., 2008)

***Response to Comment R2-15:*** *Apologies for including Taurano in the wrong table. Agree. Taurano eruption consists of several fall deposits occurred between 33 and 36 cal ka BP (Di Vito et al., 2008) at Somma-Vesuvius.*

***Changes in manuscript:*** *This eruption has been moved from Table 3 to Table 2. The tables were then placed in the Supplementary materials as requested by the reviewer #1. Please see Tables S1 and S2.*